# Recurrent Self-Attention Dynamics:
# An Energy-Agnostic Perspective from Jacobians

**Akiyoshi Tomihari**[1,2] **Ryo Karakida**[1,3]

[1]Artificial Intelligence Research Center, AIST, Japan
[2]Department of Computer Science, The University of Tokyo, Japan
[3]RIKEN Center for Advanced Intelligence Project

tomihari@g.ecc.u-tokyo.ac.jp, karakida.ryo@aist.go.jp

## Abstract

The theoretical understanding of self-attention (SA) has been steadily progressing. A prominent line of work studies a class of SA layers that admit an energy function decreased by state updates. While it provides valuable insights into inherent biases in signal propagation, it often relies on idealized assumptions or additional constraints not necessarily present in standard SA. Thus, to broaden our understanding, this work aims to relax these energy constraints and provide an energy-agnostic characterization of inference dynamics by dynamical systems analysis. In more detail, we first consider relaxing the symmetry and single-head constraints traditionally required in energy-based formulations. Next, we show that analyzing the Jacobian matrix of the state is highly valuable when investigating more general SA architectures without necessarily admitting an energy function. It reveals that the normalization layer plays an essential role in suppressing the Lipschitzness of SA and the Jacobian's complex eigenvalues, which correspond to the oscillatory components of the dynamics. In addition, the Lyapunov exponents computed from the Jacobians demonstrate that the normalized dynamics lie close to a critical state, and this criticality serves as a strong indicator of high inference performance. Furthermore, the Jacobian perspective also enables us to develop regularization methods for training and a pseudo-energy for monitoring inference dynamics.

## 1 Introduction

The theoretical understanding of self-attention (SA), a central component of Transformer architectures [Vaswani et al., 2017], has deepened in recent years, including phenomena such as rank collapse [Noci et al., 2022] and expressive capacity [Yun et al., 2020, Kajitsuka and Sato, 2024, Kingma and Ba, 2025]. One major line of research formulates attention mechanisms as processes that minimize explicit or implicit energy functions [Ramsauer et al., 2021, Yang et al., 2022, Hoover et al., 2023]. Since these energy functions serve as potential functions for gradient flows or as Lyapunov functions, they offer convergence guarantees and provide intuitive explanations for behaviors such as clustering and rank collapse in recurrent SA architectures [Geshkovski et al., 2025, 2023, 2024, Bruno et al., 2025]. They restrict attention dynamics to a hypersphere, typically as a result of normalization. This facilitates theoretical analysis and can yield well-behaved functional properties [Castin et al., 2024].

However, energy-based formulations often rely on idealized assumptions and require architectural modifications. These include imposing constraints on the weight matrices, limiting the number

of attention heads to one, and modeling state updates in the continuum limit. In addition, some architectures inspired by Hopfield networks replace SA with cross-attention [Ramsauer et al., 2021] or require double softmax passes [Hoover et al., 2023, Hu et al., 2025]. Although they can be effective for exploring a frontier of new architectures, their utility still remains limited for quantitatively understanding or improving existing, realistic SA models.

In this work, we deepen the understanding of SA by extending energy-based analysis and employing a more general stability analysis from a dynamical systems perspective.

- First, we revisit the energy-based formulation and partially relax traditional architectural constraints, such as symmetric weights and single-head assumptions, to better approximate realistic SA settings (Section 4). These relaxed constraints provide insights into designing regularization methods, which we experimentally explore later in Section 6.2.

- To study a broader class of SAs more flexibly, we next analyze the Jacobian matrix of SA dynamics (Section 5). The Jacobian approach is more general than the energy-based analysis (a.k.a. Lyapunov's direct method) in the sense that it characterizes linear stability (a.k.a. Lyapunov's indirect method) and enables us to more easily detect non-stationary dynamics, including oscillations. We demonstrate that normalization layers, unique to discrete updates, play a critical role in stabilizing dynamics. Specifically, they effectively suppress the Jacobian's spectral norm (Proposition 5.1) and control oscillatory behaviors by normalizing the complex eigenvalues of the Jacobian (Section 5.2). In addition to the understanding of the normalization role, we empirically reveal that high-performance SA models exhibit a maximum Lyapunov exponent close to zero, suggesting that rich non-stationary inference dynamics emerge at the boundary between convergence and instability.

- Finally, we investigate test-time scaling (performance improvement as the number of iterations increases) through the lens of Jacobians. We show that regularizing the spectral norm of weight matrices in SA improves performance (Section 6.2), and that the Jacobian offers an interpretation of the pseudo-energy proposed in prior work, linking it to large eigenvalues (Section 6.3).

Thus, our work broadens the dynamical understanding of SA and highlights the usefulness of the Jacobians and the Lyapunov exponent as promising and fundamental tools for further exploration of realistic SA architectures.

## 2 Related work

**Energy-based understanding.** Geshkovski et al. [2025, 2023, 2024], Karagodin et al. [2024], Bruno et al. [2025] formulated recurrent SA dynamics as interactions among tokens ("particles"), enabling theoretical analysis of phenomena such as meta-stable clustering and rank-one collapse. Their continuous-time dynamics monotonically decrease an energy (Lyapunov) function, typically requiring constraints like single-head attention, hyperspherical token states, and symmetric weights. Yang et al. [2022] similarly interpreted Transformers as alternating minimization of energy functions, though with stricter conditions on step sizes and fixed-point proximity. Ramsauer et al. [2021] formalized the attention mechanism as modern Hopfield networks, and Hoover et al. [2023], Hu et al. [2025] further developed energy functions for Transformers. We do not address approaches based on the Hopfield networks, as they require architectural modifications distinct from standard self-attention.

**Jacobian-based analysis.** The Jacobian of state updates is fundamental for characterizing neural network dynamics. For example, it has been used to analyze edge-of-chaos behavior for stable signal propagation and gradient control [Boedecker et al., 2012, Poole et al., 2016, Pennington et al., 2017], and to investigate discrete-time stability in dynamics with anti-symmetric matrices [Haber and Ruthotto, 2017, Chang et al., 2019]. Several studies have explored Jacobian-based regularization, including for generalization [Yoshida and Miyato, 2017] and for continual learning [Lewandowski et al., 2025]. Regarding SA specifically, Noci et al. [2022] analyzed Jacobians to explain rank collapse, while Castin et al. [2024] evaluated their spectral properties mathematically. In this work, we use Jacobian analysis to understand inference dynamics in realistic SAs and also employ them as regularizers and performance indicators.

**Looped architectures.** Looped architectures in Transformers have been studied since their introduction by Dehghani et al. [2018]. Yang et al. [2024], Giannou et al. [2023] showed that looped Transformers can learn algorithmic tasks, and Saunshi et al. [2025] further demonstrated their effectiveness in enhancing reasoning via strong inductive bias. Geiping et al. [2025] increased the number of loop iterations to improve performance on reasoning benchmarks, and Bansal et al. [2022] showed that looped architectures generalize to harder problems. Miyato et al. [2025] proposed artificial Kuramoto oscillatory neurons (AKOrN), a looped architecture that successfully solves tasks in a neuroscience-inspired manner, demonstrating strong empirical results in unsupervised object discovery, adversarial robustness, calibrated uncertainty quantification, and reasoning. Weight tying in ALBERT [Lan et al., 2020] and fixed-point computation in equilibrium models [Bai et al., 2019] are also interpreted as looped architectures. For a more detailed overview of previous work, see the extended related work in Section C.

## 3 Preliminaries

**Notations.** For a matrix $A$, we use the subscripts $A_{[i,j]}$, $A_{[i,:]}$, and $A_{[:,j]}$ to denote the $(i,j)$-th entry, the $i$-th row, and the $j$-th column of $A$, respectively. We denote the time index by $X^{(t)}$ and the head index in multi-head attention by $W_h$. All derivatives are computed using the numerator layout.

**Self-attention.** Multi-head self-attention (MSA) is defined as

$$\text{MSA}(X) := \text{Concat}(\text{SA}_1(X), \ldots, \text{SA}_H(X)) W^O, \tag{1}$$

and each SA head $\text{SA}_h(X)$ is defined as

$$\text{SA}_h(X) := \text{softmax}(\beta X W_h^Q W_h^{K\top} X^\top) X W_h^V, \tag{2}$$

for $h = 1, \ldots, H$. Here, $X \in \mathbb{R}^{S \times D}$ denotes a sequence of $S$ tokens, each represented by a $D$-dimensional embedding. The weight matrices $W_h^Q, W_h^K, W_h^V \in \mathbb{R}^{D \times D_H}$ correspond to the query, key, and value projections for head $h$, and $W^O \in \mathbb{R}^{D \times D}$ is the output projection matrix. Typically, the head dimension and scaling factor are set to $D_H = D/H$ and $\beta = 1/\sqrt{D_H}$.

**Self-attention with energy functions.** Geshkovski et al. [2025], Karagodin et al. [2024] used continuous equations and particle interpretation of tokens to model state-update dynamics of SA as:

$$\dot{X} = \text{Proj}_X \left( \text{softmax}(\beta X W^Q W^{K\top} X^\top) X W^V \right) \tag{3}$$

To have an energy function, the previous work has assumed constraints such as

$$W^Q W^{K\top} = W^V = W^{V\top} \quad \text{or} \quad W^Q = W^K = W^V = I_D, \tag{4}$$

depending on the analyses. [Bruno et al., 2025] further assumes an unnormalized version of softmax. Under these conditions, the SA update can decrease an energy function. That is, the dynamics evolve in a way that monotonically decreases the energy, thereby ensuring the Lyapunov stability. Because these models suppose symmetric weights, we refer to a class of SA layers with symmetric weights and Lyapunov functions as **symmetric SA**. Yang et al. [2022] also formalized updates of SA using a symmetric matrix (Appendix C).

**Spherical constraint.** To facilitate theoretical analysis of SA, several studies [Geshkovski et al., 2025, Miyato et al., 2025] have introduced a spherical constraint on token vectors by enforcing that each token vector has unit norm. This constraint enables an interpretation of token interactions as dynamics of particles on a hypersphere, and plays a key role in controlling the Lipschitz continuity of SA [Castin et al., 2024]. There are two commonly used operators with a spherical constraint: a normalization operator $\Pi$ that enforces the spherical constraint, and a projection operator $\text{Proj}$ that projects onto the tangent space of the sphere. Given a token matrix $X, Y \in \mathbb{R}^{S \times D}$ such that $\|X_{[i,:]}\| = 1$, these operators are defined token-wisely as:

$$\Pi(Y)_{[i,:]} = Y_{[i,:]}/\|Y_{[i,:]}\|, \quad \text{Proj}_X(Y)_{[i,:]} = \left( I_D - X_{[i,:]} X_{[i,:]}^\top \right) Y_{[i,:]}. \tag{5}$$

Here, $\Pi(Y)$ projects each token vector $Y_{[i,:]}$ onto the unit hypersphere. $\text{Proj}_X(Y)$ projects $Y_{[i,:]}$ orthogonally to $X_{[i,:]}$, restricting updates to the tangent space of the sphere.

In practical Transformer architectures, the spherical normalization can be interpreted as a special case of Root Mean Square Normalization (RMSNorm) [Zhang and Sennrich, 2019], which is applied to the input matrix $\boldsymbol{Y} \in \mathbb{R}^{S \times D}$ as:

$$\text{RMSNorm}(\boldsymbol{Y})_{[i,:]} = \text{diag}(\boldsymbol{\gamma})\Pi(\boldsymbol{Y})_{[i,:]}, \tag{6}$$

where $\boldsymbol{\gamma} \in \mathbb{R}^D$ is a trainable parameter vector. RMSNorm rescales each token to have unit norm and applies element-wise scaling using the learned parameter $\boldsymbol{\gamma}$, while $\Pi$ can be interpreted as the special case with $\boldsymbol{\gamma} = \mathbf{1}$. As we will show in Section 5, the trainable parameter $\boldsymbol{\gamma}$ plays an important role in stabilizing Jacobians.

**AKOrN.** AKOrN [Miyato et al., 2025] integrates a generalized Kuramoto model into an artificial neural network by updating oscillatory neurons through a looped structure. The connectivity among oscillators is implemented in several ways. In this work, we focus on one of their AKOrNs that uses SA. Given a sequential input $\boldsymbol{C} \in \mathbb{R}^{S \times D}$, AKOrN initializes $\boldsymbol{X} \in \mathbb{R}^{S \times D}$ using $\boldsymbol{C}$. Each token vector $\boldsymbol{X}_{[i,:]}$ ($i = 1 \cdots S$) is partitioned into $N$-dimensional vectors, referred to as the oscillators. AKOrN iteratively updates states using a **Kuramoto layer** as follows:

$$\Delta \boldsymbol{X}^{(t)} = \text{Omg}^{(\text{osc})}(\boldsymbol{X}^{(t)}) + \text{Proj}_{\boldsymbol{X}^{(t)}}^{(\text{osc})}\left(\boldsymbol{C} + \text{MSA}(\boldsymbol{X}^{(t)})\right), \tag{7}$$

$$\boldsymbol{X}^{(t+1)} = \Pi^{(\text{osc})}\left(\boldsymbol{X}^{(t)} + \eta \Delta \boldsymbol{X}^{(t)}\right), \tag{8}$$

where $\eta$ denotes a positive discrete step size. The Omega layer (Omg) is given by a linear transformation by anti-symmetric matrices and determines the rotational dynamics of oscillators. The projection operator $\text{Proj}_{\boldsymbol{X}}(\boldsymbol{Y})$ and the normalization operator $\Pi(\boldsymbol{Y})$ are applied independently to each oscillator. We use the notation $\bullet^{(\text{osc})}$ to denote oscillator-wise operations. We provide further details of AKOrN in Appendix C.

Although the existence of an energy function can be guaranteed under certain special conditions for Kuramoto models, practical implementations of the Kuramoto layer do not assume such conditions in order to achieve better performance.

**Iterative self-attention.** Previous studies on looped architectures [Saunshi et al., 2025, Bansal et al., 2022] have shown that injecting the input $\boldsymbol{C} \in \mathbb{R}^{S \times D}$ into the loop is important for achieving test-time scaling. As we show later, our theory (Proposition 5.1) indicates that the normalization layer plays a critical role in controlling the norm of the Jacobian matrix, particularly in the case of RMS normalization, which is widely used in practice. Based on these insights, we propose and investigate the following update rule, referred to as **iterative self-attention (ItrSA)**:

$$\Delta \boldsymbol{X}^{(t)} = \boldsymbol{C} + \text{MSA}(\boldsymbol{X}^{(t)}), \quad \boldsymbol{X}^{(t+1)} = \text{RMSNorm}\left(\boldsymbol{X}^{(t)} + \eta \Delta \boldsymbol{X}^{(t)}\right). \tag{9}$$

Since ItrSA does not involve oscillator-wise operations and the Omg layer, both of which are distinctive components of AKOrN, it is suitable as a baseline to compare against energy-based SAs.

## 4 Energy-based analysis

As described in the previous section, energy-based symmetric SA involves three constraints: (i) the symmetry of weight matrices, (ii) a single head, and (iii) a continuous-time limit. To accommodate more realistic SA architectures, we partially relax (i) and (ii) in the following.

**Extension of weight symmetry.** Symmetric SA imposes symmetric constraints on both $\boldsymbol{W}^Q \boldsymbol{W}^{K\top}$ and $\boldsymbol{W}^V$. We relax these constraints in the following proposition.

**Proposition 4.1.** *Consider the continuous-time dynamics for single-head SA equipped with projection (3). The energy function*

$$E_{single}(\boldsymbol{X}) = -\sum_{i,j} \exp\left(\beta \boldsymbol{X}_{[i,:]}^\top \boldsymbol{W}^Q \boldsymbol{W}^{K\top} \boldsymbol{X}_{[j,:]}\right) \tag{10}$$

*is monotonically decreasing as $dE_{single}(\boldsymbol{X})/dt \leq 0$ under the condition:*

$$\boldsymbol{W}^V = (\boldsymbol{W}^K \boldsymbol{W}^{Q\top} + \boldsymbol{W}^Q \boldsymbol{W}^{K\top})/2. \tag{11}$$

**Multi-head energy.** Although energy functions have been proposed for single-head SA, no corresponding formulation exists for MSA, which is commonly used in practice. We extend the above result to the multi-head setting as follows.

**Proposition 4.2.** *Consider the continuous-time dynamics for multi-head SA without projection:* $d\boldsymbol{X}/dt = \sum_{h=1}^{H} SA_h(\boldsymbol{X})$. *An energy function*

$$E_{multi}(\boldsymbol{X}) = -\sum_{h} \sum_{i,j} \exp\left(\beta \boldsymbol{X}_{[i,:]}^{\top} \boldsymbol{W}_h^Q \boldsymbol{W}_h^{K\top} \boldsymbol{X}_{[j,:]}\right) \tag{12}$$

*is monotonically decreasing as* $dE_{multi}(\boldsymbol{X})/dt \leq 0$ *under the condition*

$$\boldsymbol{W}_h^V = (\boldsymbol{W}_h^K \boldsymbol{W}_h^{Q\top} + \boldsymbol{W}_h^Q \boldsymbol{W}_h^{K\top})/2, \quad \boldsymbol{W}_h^Q \boldsymbol{W}_h^{K\top} = \boldsymbol{U}_{1,h}\boldsymbol{U}_{2,h}^{\top}, \tag{13}$$

*where* $\boldsymbol{U}_{1(2),h} \in \mathbb{R}^{D \times D/(2H)}$ $(h \in [1, H])$ *satisfies the orthogonality condition* $\boldsymbol{U}_{k,h}^{\top}\boldsymbol{U}_{k',h'} = \delta_{hh'}\delta_{kk'}\boldsymbol{I}_{D/(2H)}$.

Propositions 4.1 and 4.2 imply that certain structures of weight matrices are desirable to ensure the existence of an energy function. Specifically, $W_h^Q W_h^{K\top}$ can be asymmetric, whereas $W_h^V$ should remain symmetric. In the multi-head scenario, a low-rank structure in the QK product is required. This aligns with practical Transformers, as they typically exhibit a low-rank structure due to the small inner dimension (the width of $W_h^Q, W_h^K$). We refer to architectures that incorporate these properties as **generalized symmetric SA**, and we will explore their effectiveness in our experiments (Section 6.2). The proofs are provided in Appendices A.2 and A.3.

# 5 Jacobian-based analysis

In general, energy functions are used to guarantee the convergence of dynamics to fixed points (a.k.a. Lyapunov's direct method) [Khalil, 2002]. While this is a concrete approach to achieving stable dynamics, the construction of energy functions is usually unsystematic, and thus it is not obvious whether we can handle more realistic SA dynamics (e.g., discrete updates with normalization). Furthermore, recent experimental results have reported non-stationary dynamics (e.g., oscillations) [Karagodin et al., 2024, Miyato et al., 2025], suggesting the need for more flexible approaches applicable to richer dynamics. Thus, we turn to analyzing the Jacobian matrix of state updates. The Jacobian controls the Lipschitzness of the function and also naturally appears in linear stability analysis (a.k.a. Lyapunov's indirect method), where state updates are locally described by $\boldsymbol{f}(\boldsymbol{x} + \Delta\boldsymbol{x}) \approx \boldsymbol{f}(\boldsymbol{x}) + \boldsymbol{J}(\boldsymbol{x}_t)\Delta\boldsymbol{x}$ with the Jacobian $\boldsymbol{J} := \partial\boldsymbol{f}/\partial\boldsymbol{x}$.

## 5.1 Normalization and spectral norm

Normalization operators, which do not appear in continuous-time dynamics, are essential in the discrete setting because discretizing state updates causes the state vector to deviate from the hypersphere. We find that the normalization operators suppress the Jacobian's eigenvalues.

**Proposition 5.1.** *Suppose that, in the update of ItrSA (9), the input to the normalization layer satisfies* $\|\boldsymbol{X}_{[i,:]} + \eta\Delta\boldsymbol{X}_{[i,:]}\| \geq R$ *for all* $i \in [1, S]$. *Then, the spectral norm of the Jacobian satisfies*

$$\left\|\frac{\partial\,\mathrm{RMSNorm}(\boldsymbol{X} + \eta\Delta\boldsymbol{X})}{\partial\boldsymbol{X}}\right\|_2 \leq \frac{\max_j(|\gamma_j|)}{R}\left(1 + \eta\,\|\boldsymbol{J}_{MSA}(\boldsymbol{X})\|_2\right), \tag{14}$$

*where* $\boldsymbol{J}_{MSA}(\boldsymbol{X}) := \partial\,\mathrm{MSA}(\boldsymbol{X})/\partial\boldsymbol{X}$ *denotes the Jacobian of* $\mathrm{MSA}$.

We show the proof in Appendix A.4. This proposition highlights the key stabilizing effect of normalization: the spectral norm is inversely proportional to $R$. This effect appears to be particularly significant for preventing signal explosion in looped architectures, where the same operation is repeatedly applied.

Figure 2a shows that, in a practical model, normalization reduces the maximum Lyapunov exponent. This exponent corresponds to a time-averaged maximum singular value of $\boldsymbol{J}$, which is further explained and analyzed in Section 5.3. This result supports the stabilizing effect of normalization implied by Proposition 5.1.

In addition, under the assumption that $\|\Delta \boldsymbol{X}_{[i,:]}\| \geq \varepsilon$ for some constant $\varepsilon > 0$ for all $i$, we can further show that as $\eta \to \infty$,

$$\left\| \frac{\partial \operatorname{RMSNorm}(\boldsymbol{X} + \eta \Delta \boldsymbol{X})}{\partial \boldsymbol{X}} \right\|_2 = O(1). \tag{15}$$

That is, the Jacobian norm remains bounded even for a large $\eta$ (see Appendix A.4).

**Jacobian eigenvalues of SA.** Castin et al. [2024] provide an upper bound on the maximum eigenvalue (in the form of a Lipschitz constant) of the Jacobian of SA defined as $\boldsymbol{J}_{\mathrm{MSA}}(\boldsymbol{X}) \coloneqq \partial \operatorname{MSA}(\boldsymbol{X})/\partial \boldsymbol{X}$. Specifically, their results in Theorem 3.3 and Lemma 3.8 state that the Jacobian $\boldsymbol{J}_{\mathrm{MSA}}(\boldsymbol{X})$ satisfies for input tokens $\boldsymbol{X} \in \mathbb{R}^{S \times D}$ such that $\|\boldsymbol{X}_{[i,:]}\| \leq r$ for all $i \in [1, S]$.

$$\|\boldsymbol{J}_{\mathrm{MSA}}(\boldsymbol{X})\|_2 \leq \sum_{h=1}^{H} \sqrt{3} \|\boldsymbol{W}_h^O\|_2 \|\boldsymbol{W}_h^V\|_2 \sqrt{\|\beta \boldsymbol{W}_h^Q \boldsymbol{W}_h^{K\top}\|_2 r^4 (S+1) + S}, \tag{16}$$

This inequality indicates that the right-hand side of Eq. (14) is bounded independently of the input $\boldsymbol{X}$. It further implies that when the norms of the weight matrices $\|\boldsymbol{W}_h^O\|_2, \|\boldsymbol{W}_h^V\|_2$, or $\|\boldsymbol{W}_h^Q \boldsymbol{W}_h^{K\top}\|_2$, or the number of tokens $S$ becomes large, the spectral norm can also become large. This is precisely the issue that normalization techniques can address. Figure 1 demonstrates that the spectral norm of the untrained SA's Jacobian can be effectively reduced through normalization (see Appendix B.4 for experimental settings). Interestingly, we empirically observed that the spectral norm is not only reduced by normalization but remains $O(1)$ with respect to the number of tokens. This suggests that the current theoretical bound (Eqs. (14) and (16)) is conservative, and a tighter bound remains future theoretical work.

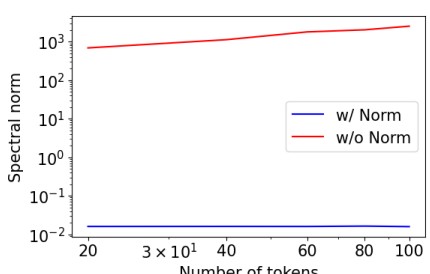

Figure 1: **Normalization improves the spectral norm of SA's Jacobian.**

## 5.2 Normalization of oscillatory components

To clarify differences between continuous and discrete updates and highlight the role of normalization, we analyze the following simplified dynamics and their associated Jacobians:

$$\dot{\boldsymbol{x}} = \boldsymbol{\Omega} \boldsymbol{x}, \qquad \boldsymbol{J}(\boldsymbol{x}) = \boldsymbol{\Omega} \qquad \text{(continuous)} \tag{17}$$

$$\boldsymbol{x}^{(t+1)} = (\boldsymbol{I}_D + \eta \boldsymbol{\Omega}) \boldsymbol{x}^{(t)}, \qquad \boldsymbol{J}(\boldsymbol{x}^{(t)}) = \boldsymbol{I}_D + \eta \boldsymbol{\Omega} \qquad \text{(discrete w/o Norm)} \tag{18}$$

$$\boldsymbol{x}^{(t+1)} = \Pi\big((\boldsymbol{I}_D + \eta \boldsymbol{\Omega}) \boldsymbol{x}^{(t)}\big), \quad \boldsymbol{J}(\boldsymbol{x}^{(t)}) = \left(\boldsymbol{I}_D - \frac{\boldsymbol{y}\boldsymbol{y}^\top}{\|\boldsymbol{y}\|^2}\right) \frac{\boldsymbol{I}_D + \eta \boldsymbol{\Omega}}{\|\boldsymbol{y}\|} \quad \text{(discrete w/ Norm)} \tag{19}$$

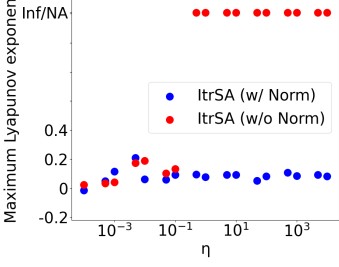
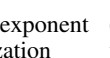
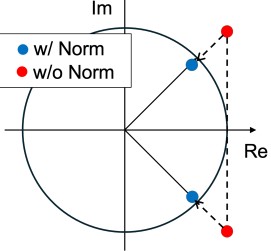
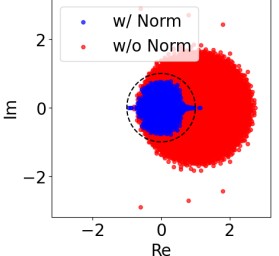

(a) Maximum Lyapunov exponent with and without normalization

(b) Effect of normalization on eigenvalues in oscillatory case

(c) Eigenvalue distribution in the complex plane

Figure 2: **Normalization layers play a crucial role in controlling the Jacobian's eigenvalues.** (a) and (c) show results on the Sudoku dataset.

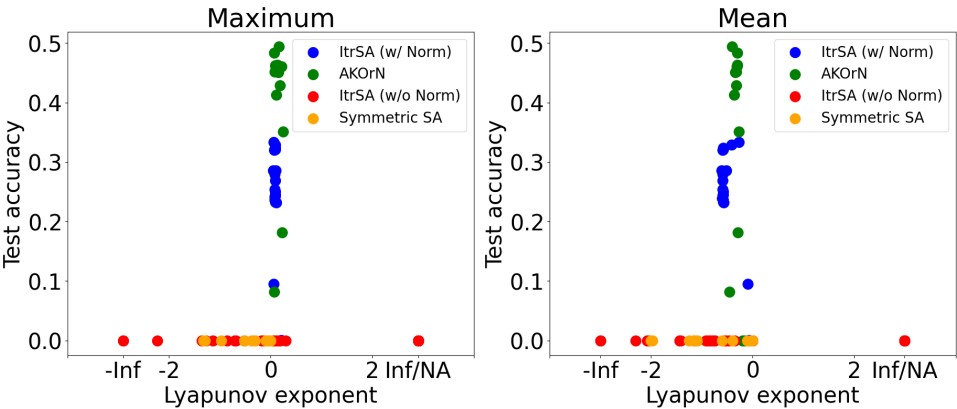

Figure 3: **Normalization drives the Lyapunov exponents toward zero and enables high accuracy.**

where $\boldsymbol{\Omega}$ is an anti-symmetric matrix and $\boldsymbol{y} = (\boldsymbol{I}_D + \eta\boldsymbol{\Omega})\boldsymbol{x}^{(t)}$. Since an anti-symmetric matrix has purely imaginary eigenvalues, these dynamics represent simple oscillatory systems. The discrete dynamics with normalization can also be interpreted as isolating the Omega layer used in AKOrN.

It is known that the pure imaginary eigenvalues in the continuous-time limit are essential for capturing long-term signal dependencies, but can be significantly damaged by discretization [Chang et al., 2019]. Generally, in continuous-time systems $\dot{\boldsymbol{x}} = \boldsymbol{f}(\boldsymbol{x})$, the equilibrium point is Lyapunov stable if all eigenvalues $\lambda_j$ of the Jacobian $\boldsymbol{J}(\boldsymbol{x})$ satisfy $\mathrm{Re}(\lambda_j) \leq 0$. In Eq. (17), the Jacobian $\boldsymbol{J}(\boldsymbol{x}) = \boldsymbol{\Omega}$ has purely imaginary eigenvalues and they are on the boundary of stability, allowing persistent oscillations. In contrast, the equilibrium points of discrete-time systems $\boldsymbol{x}^{(t+1)} = \boldsymbol{f}(\boldsymbol{x}^{(t)})$ are Lyapunov stable if all eigenvalues $\lambda_j$ of the Jacobian satisfy $|\lambda_j| \leq 1$. For Eq. (18), all eigenvalues of the Jacobian $\boldsymbol{J}(\boldsymbol{x}) = \boldsymbol{I}_D + \eta\boldsymbol{\Omega}$ take the form $1 \pm i\eta\omega_j$ for $\omega_j \geq 0$, implying that $|\lambda_j| \geq 1$. Therefore, the system becomes unstable. To avoid the fundamental instability arising from discretization, previous work on architecture design inspired by dynamical systems proposed to add a diffusion term to the anti-symmetric weight matrix, i.e., $\boldsymbol{\Omega} - \gamma\boldsymbol{I}$ ($\gamma > 0$) [Haber and Ruthotto, 2017, Chang et al., 2019].

We find that a normalization layer (19) serves as an alternative way to mitigate this instability by effectively rescaling the system through division by $\|\boldsymbol{y}\|$. For simplicity, suppose that all eigenvalues of $\Omega$ degenerate to the same $\omega_j = \omega$. After a straightforward calculation, we obtain $|\lambda_j| \leq 1$ (see Appendix A.5 for the derivation). The effect of this normalization is illustrated in Figure 2b. Although this scenario represents an idealized setting, the normalization of imaginary components is empirically observed even in the case of SA, as shown in Figure 2.

### 5.3 Lyapunov exponent indicates criticality

The Lyapunov exponent measures the exponential rate at which trajectories locally converge or diverge in dynamical systems [Khalil, 2002] (see details in Appendix B.3). Intuitively, it corresponds to the time-averaged value of $\ln \sigma_i(\boldsymbol{J})$, where $\sigma_i(\boldsymbol{J})$ denotes the Jacobian's singular values. A positive exponent indicates instability and sensitivity to initial conditions, whereas a negative exponent implies convergence. An exponent close to zero characterizes a critical regime, often referred to as the edge of chaos, where signals neither explode nor vanish and can propagate for a long period. This critical regime has been reported to correlate with the high performance of neural networks across various contexts [Boedecker et al., 2012, Poole et al., 2016, Pennington et al., 2017].

Figure 3 shows that SA models achieving high test accuracy empirically have Lyapunov exponents close to zero, thus operating near criticality. The maximum and mean exponents display nearly identical behaviors. We vary hyper-parameters, including $\eta$ and the norm of weight matrices, and plot multiple points (see details in Appendix B.3). In models with normalization layers, the exponent tends to concentrate around zero, indicating the criticality. High accuracy is achieved only in these models. This supports the stabilizing effect of normalization implied by Proposition 5.1 and Section 5.2. In contrast, energy-based symmetric SA models show negative exponents, consistent with their Lyapunov stability, leading to lower accuracy. Interestingly, the maximum Lyapunov exponent of successful models is slightly positive ($\sim 0.1$), indicating that the dynamical state resides near

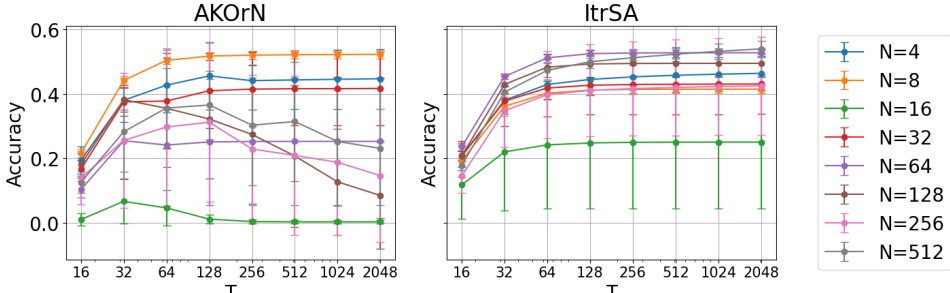

Figure 4: **ItrSA consistently improves accuracy as the number of loops $T$ increases, regardless of the value of oscillator dimension $N$.** Note that $N = 512$ corresponds to the setting where tokens are not split into oscillators. Error bars indicate the standard deviation.

criticality from the chaotic side. Notably, we observed that the dynamics with this slightly positive maximum Lyapunov exponent indicate the sensitivity to initial conditions, implying chaotic behavior (see Figure S.5 in the appendix). This observation aligns with previous reports of positive maximum Lyapunov exponents in some Transformers [Inoue et al., 2022, Liu et al., 2024, Tong et al., 2025]. We observed similar Lyapunov exponents across the CIFAR-10 dataset (Figure S.7) and the language modeling task (Table S.4). We further observed that as the number of attention heads increases, the Lyapunov exponents tend to increase (Figures S.1 and S.2). This implies that multi-head attention would favor a more dynamic state.

## 6 Quantitative insight into inference dynamics

Here, we experimentally investigate the test-time scaling of inference in looped architectures. We mainly focus on evaluation on the Sudoku task using the SATNet [Wang et al., 2019] dataset for in-distribution (ID) data and the RRN dataset [Palm et al., 2018] for out-of-distribution (OOD) data. At test time, we increased the number of loops beyond the training setting of $T = 16$. Details are provided in Appendix B.

### 6.1 Test-time scaling and normalization

Miyato et al. [2025] showed that AKOrN exhibits test-time scaling, whereas ItrSA does not, suggesting the superiority of AKOrN over ItrSA. However, with our formulation of ItrSA, Figure 4 demonstrates that ItrSA also exhibits test-time scaling. Moreover, it shows that AKOrN fails to maintain test-time scaling when $N$ becomes large, which is consistent with the observations by Miyato et al. [2025]. This issue can be mitigated by applying RMS normalization, where we use oscillator-wise RMS normalization. The learned scaling parameter $\gamma$ prevents the Jacobian from becoming excessively large. Empirically, we confirmed that the trained $\gamma$ remains small (see Table S.3), eliminating the need for explicit clipping such as $|\gamma_i| \leq 1$.

### 6.2 Application to regularization

To apply our energy-based and Jacobian-based analysis, we add the following regularization term $R$, scaled by a tunable parameter $\lambda$.

**Method (i): Energy-based regularization.** Proposition 4.2 suggests that SA architectures satisfying specific conditions inherently minimize an energy function. To investigate the practical utility of this property, we consider applying the constraints identified in the proposition as a regularization term to the existing multi-head MSA. We add the following energy-based regularization term to the loss function during the training of ItrSA, which can be interpreted as an approximation of energy-based SA models. By defining the concatenation as $\boldsymbol{W}^V := [\boldsymbol{W}_1^V, \cdots, \boldsymbol{W}_H^V] \in \mathbb{R}^{D \times D}$, we introduce

$$R_{\text{E-multi}} := \left\| \boldsymbol{W}^V \boldsymbol{W}^O - (\boldsymbol{W}^V \boldsymbol{W}^O)^\top \right\|_F^2, \tag{20}$$

where both $\boldsymbol{W}^V$ and $\boldsymbol{W}^O$ are implemented as **orthogonal matrices**. Note that each $\boldsymbol{W}_h^V$ in the proposition is interpreted as the product $\boldsymbol{W}_h^V \boldsymbol{W}_h^O$ in ItrSA. If $R_{\text{E-multi}} = 0$, $\boldsymbol{W}_h^V \boldsymbol{W}_h^O$ becomes

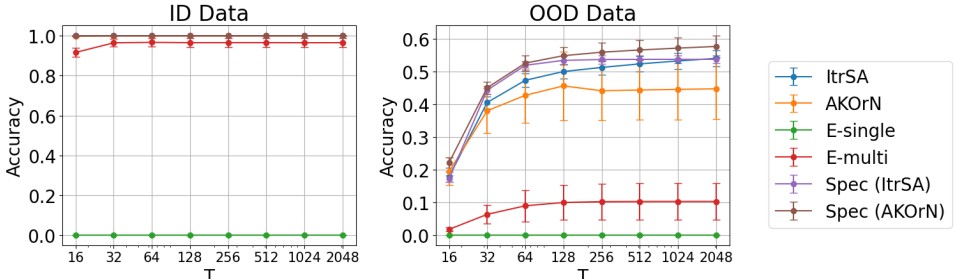

Figure 5: **Energy-based regularization ("E-single" and "E-multi") underperforms the original methods, while Jacobian spectral regularization ("Spec") outperforms.** We set $H = 8$ except for E-single ($H = 1$). For AKOrN, we used $N = 4$.

symmetric as implied by Proposition 4.2. For a single-head case, we can also propose Proposition 4.1 as a regularization term $R_{\text{E-single}}$ (see Appendix B.2).

**Method (ii): Jacobian spectral regularization.** On the other hand, controlling the Jacobian spectra is an effective way to stabilize neural networks. Following the regularization proposed by Lewandowski et al. [2025], we introduce the following regularization term:

$$R_{\text{Spec}} = \sum_{\boldsymbol{W} \in \text{SA}} \left( \sigma^2(\boldsymbol{W}) - 1 \right)^2 + \sum_{\boldsymbol{b} \in \text{SA}} \|\boldsymbol{b}\|_2^4, \tag{21}$$

where the summations are taken over all weight matrices $\boldsymbol{W}$ and bias terms $\boldsymbol{b}$ in the SA modules, and $\sigma(\boldsymbol{W})$ denotes the largest singular value of $\boldsymbol{W}$. This regularization encourages the singular values to be close to 1, which has been shown to be beneficial for recursive architectures [Chang et al., 2019]. We apply $R_{\text{Spec}}$ to both ItrSA and AKOrN.

**Limitation of energy-based regularization.** Figure 5 shows the effects of the regularization methods. The accuracy of E-multi is lower than that without regularization. The Lyapunov exponents shown in Figure S.2 indicates that multi-head energy regularization encourages more convergent dynamics and this does not necessarily yield better performance. E-single fails to reduce the training loss and fails even on ID tasks possibly due to the single-head constraint. These results suggest that energy-based regularization may be unnecessary, casting doubt on the validity of the energy-based perspective for practical applications.

**Spectral regularization is particularly effective in AKOrN.** Spectral regularization substantially enhances the performance of both AKOrN and ItrSA, and the effect is especially pronounced in AKOrN. Equation (16) shows that the maximum eigenvalue of the Jacobian can grow significantly when the weight matrices have large norms. Spectral regularization addresses this issue by directly constraining the Jacobian spectrum. When $N = 8$, AKOrN achieves the best performance and spectral regularization is also effective (Figure S.3).

### 6.3 An interpretation of pseudo-energy via Jacobian

While it is non-trivial for general SA dynamics to have an energy function, Miyato et al. [2025] empirically found that AKOrN approximately decreases the quantity $E_{\text{pseudo}}(t) \coloneqq -\operatorname{Tr}(\boldsymbol{X}^{(t)\top}\boldsymbol{Y}^{(t)})$, which we refer to as pseudo-energy, during the updates of Eqs. (7) and (8), where $\boldsymbol{Y}^{(t)} = \boldsymbol{C} + \text{MSA}(\boldsymbol{X}^{(t)})$. They also reported that the prediction with a lower pseudo-energy performs better. Under certain symmetry assumptions without SA, it reduces to the energy function of a generalized Kuramoto model. However, its interpretation in AKOrN with SA remains unclear.

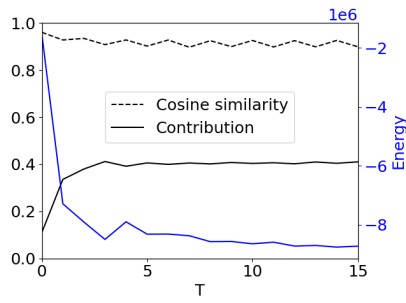

Figure 6: Test-time inference of AKOrN on the Sudoku dataset.

We found that the Jacobian provides an interpretation. Suppose that $\text{vec}(\text{MSA}(\boldsymbol{X}^{(t)})) \in \mathbb{R}^{DS}$ is well-approximated

by $\boldsymbol{J}_t \boldsymbol{x}_t$, where $\boldsymbol{x}_t \in \mathbb{R}^{DS}$ denotes $\text{vec}(\boldsymbol{X}^{(t)})$ and $\boldsymbol{J}_t = \partial\, \text{MSA}(\boldsymbol{X}^{(t)})/\partial \boldsymbol{X}^{(t)} \in \mathbb{R}^{DS \times DS}$. Figure 6 shows that while $E_{\text{pseudo}}$ significantly decreases, the cosine similarity between $\text{vec}(\text{MSA}(\boldsymbol{X}^{(t)}))$ and $\boldsymbol{J}_t \boldsymbol{x}_t$ remains high throughout iterations. Under this approximation, neglecting a constant term $\boldsymbol{C}$, we obtain the relation $E_{\text{pseudo}} = -\boldsymbol{x}_t^\top \boldsymbol{S}_t \boldsymbol{x}_t/2$, a quadratic form involving a symmetric matrix $\boldsymbol{S}_t = \boldsymbol{J}_t + \boldsymbol{J}_t^\top$. We further expand $\boldsymbol{x}_t$ in the orthonormal basis of $\boldsymbol{S}_t$ as $\boldsymbol{x}_t = \sum_{k=1}^{DS} a_t^k \boldsymbol{v}_t^k$, and the eigenvectors are sorted in descending order of their eigenvalues ($\lambda_t^1 \geq \cdots \geq \lambda_t^{DS}$). The *contribution index* in the figure quantifies the extent to which the state $\boldsymbol{x}_t$ is captured by the top $2\%$ eigenvalues, specifically $\sum_{k \leq 0.02DS} (a_t^k)^2 / \sum_{k=1}^{DS} (a_t^k)^2$. During inference, the proportion of the state in the top $2\%$ eigenspace increases monotonically, eventually dominating nearly $40\%$. This suggests that the loop effectively performs a computation analogous to power iteration, but constrained to real space with positive eigenvalues. Thus, the Jacobian provides a meaningful and promising interpretation for the observed decrease of pseudo-energy during AKOrN inference.

As a side note, in Section B.5, a more detailed analysis of the Jacobian matrix implies that this alignment to the eigenspace is predominantly determined by a certain time-independent matrix.

## 7 Conclusion

In this paper, we advanced the understanding and control of recurrent self-attention (SA) from a dynamical systems perspective. First, we generalized energy-based formulations to weaker symmetry and multi-head configurations closer to realistic settings. However, experiments showed that energy-regularized SA underperformed standard SA, suggesting other factors at play in practical models. Second, we analyzed the Jacobian matrix of standard SA architectures, revealing that normalization layers effectively regularize their spectral properties. We further clarify that Lyapunov exponents at criticality characterize high-performance inference, which indicates that the state update of SA is more dynamic than energy-constrained cases. We also argued how Jacobians provide quantitative insights into performance-enhancing regularization and pseudo-energy behaviors.

**Limitations.** In this work, we focused on recurrent SAs without positional encoding, masking, or MLP blocks, unlike looped Transformers in practice. Investigating how these components alter state dynamics remains an interesting direction. Regarding theoretical limitations, the upper bound in Proposition 5.1 provides a conservative upper bound that is looser than empirical observations, an issue also noted in existing analyses of SA's Jacobians without normalization. Deriving tighter bounds remains an intriguing open theoretical problem. Additionally, analytically justifying empirical phenomena, such as the Lyapunov exponent concentration (Section 5.3) and the Jacobian-based approximation (Section 6.3), by solving state dynamics would be challenging yet exciting themes for theory. We expect our findings to serve as a foundation for further theoretical and practical advancements in looped SA architectures and their rich inference dynamics.

## Acknowledgments and Disclosure of Funding

We thank the reviewers for their insightful and helpful feedback on the manuscript, and Takeru Miyato for his valuable comments. The authors acknowledge the funding support from JST FOREST (Grant No. JPMJFR226Q). RK is also supported by JSPS KAKENHI (Grant Nos. 22H05116, 23K16965).

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

# A Proofs

## A.1 Lemmas

We use the following lemmas in our derivation.

**Lemma A.1** (Singh et al.).

$$\frac{\partial \boldsymbol{AXB}}{\partial \boldsymbol{X}} = \boldsymbol{A} \otimes \boldsymbol{B}^\top \tag{22}$$

**Lemma A.2.** *Let $\boldsymbol{Y} \in \mathbb{R}^{S \times D}$ be an input matrix. Then, the Jacobian of $\Pi$ with respect to $Y$ is given by*

$$\frac{\partial \Pi(\boldsymbol{Y})}{\partial \boldsymbol{Y}} = \text{blockdiag}\left( \{ \frac{1}{\|\boldsymbol{Y}_{[i,:]}\|}(\boldsymbol{I}_D - \frac{\boldsymbol{Y}_{[i,:]}\boldsymbol{Y}_{[i,:]}^\top}{\|\boldsymbol{Y}_{[i,:]}\|^2})\}_{i=1}^S \right) \tag{23}$$

*Proof.* Since $\Pi$ operates independently on each row of $Y$, the Jacobian is block-diagonal, with each block corresponding to the derivative of a single normalized row:

$$\frac{\partial \Pi(\boldsymbol{Y})}{\partial \boldsymbol{Y}} = \text{blockdiag}(\{ \frac{\partial \Pi(\boldsymbol{Y})_{[i,:]}}{\partial \boldsymbol{Y}_{[i,:]}} \}_{i=1}^S). \tag{24}$$

For each row, we compute the gradient of the normalized vector:

$$\frac{\partial \Pi(\boldsymbol{Y})_{[i,:]}}{\partial \boldsymbol{Y}_{[i,:]}} = \frac{\partial \boldsymbol{Y}_{[i,:]}/\|\boldsymbol{Y}_{[i,:]}\|}{\partial \boldsymbol{Y}_{[i,:]}} \tag{25}$$

$$= \frac{1}{\|\boldsymbol{Y}_{[i,:]}\|}(\boldsymbol{I}_D - \frac{\boldsymbol{Y}_{[i,:]}\boldsymbol{Y}_{[i,:]}^\top}{\|\boldsymbol{Y}_{[i,:]}\|^2}). \tag{26}$$

$\square$

**Lemma A.3.** *Let $\boldsymbol{Y} \in \mathbb{R}^{S \times D}$ be an input matrix. Then, the Jacobian of $\text{RMSNorm}$ with respect to $Y$ is given by*

$$\frac{\partial \text{RMSNorm}(\boldsymbol{Y})}{\partial \boldsymbol{Y}} = \text{blockdiag}\left( \{ \frac{1}{\|\boldsymbol{Y}_{[i,:]}\|} \text{diag}(\boldsymbol{\gamma})(\boldsymbol{I}_D - \frac{\boldsymbol{Y}_{[i,:]}\boldsymbol{Y}_{[i,:]}^\top}{\|\boldsymbol{Y}_{[i,:]}\|^2})\}_{i=1}^S \right) \tag{27}$$

*Proof.* Since $\text{RMSNorm}$ is expressed as

$$\text{RMSNorm}(\boldsymbol{Y})_{[i,:]} = \text{diag}(\boldsymbol{\gamma})\Pi(\boldsymbol{Y})_{[i,:]}, \tag{28}$$

the result follows from lemma A.2. $\square$

## A.2 Proof of Proposition 4.1

**Proposition 4.1 is restated.**
*Consider the continuous-time dynamics for single-head SA equipped with projection (3). The energy function*

$$E_{single}(\boldsymbol{X}) = -\sum_{i,j} \exp\left( \beta \boldsymbol{X}_{[i,:]}^\top \boldsymbol{W}^Q \boldsymbol{W}^{K\top} \boldsymbol{X}_{[j,:]} \right) \tag{29}$$

*is monotonically decreasing as $dE_{single}(\boldsymbol{X})/dt \leq 0$ under the condition:*

$$\boldsymbol{W}^V = (\boldsymbol{W}^K \boldsymbol{W}^{Q\top} + \boldsymbol{W}^Q \boldsymbol{W}^{K\top})/2. \tag{30}$$

*Proof.* Let $\Delta = \text{softmax}(\beta \boldsymbol{X} \boldsymbol{W}^Q \boldsymbol{W}^{K\top} \boldsymbol{X}^\top) \boldsymbol{X} \boldsymbol{W}^V$ and let $\boldsymbol{A} = \boldsymbol{W}^Q \boldsymbol{W}^{K\top}$. The first-order derivative of $E_{\text{single}}(\boldsymbol{X})$ with respect to $\boldsymbol{X}_{[i,:]}$ is:

$$\frac{dE_{\text{single}}(\boldsymbol{X})}{d\boldsymbol{X}_{[i,:]}} = -\sum_{j \neq i} \frac{d\exp\left( \beta \boldsymbol{X}_{[i,:]}^\top \boldsymbol{A} \boldsymbol{X}_{[j,:]} \right)}{d\boldsymbol{X}_{[i,:]}} - \sum_{j \neq i} \frac{d\exp\left( \beta \boldsymbol{X}_{[j,:]}^\top \boldsymbol{A} \boldsymbol{X}_{[i,:]} \right)}{d\boldsymbol{X}_{[i,:]}} \tag{31}$$

$$-\frac{d\exp\left(\beta \boldsymbol{X}_{[i,:]}^\top \boldsymbol{A}\boldsymbol{X}_{[i,:]}\right)}{d\boldsymbol{X}_{[i,:]}} \tag{32}$$

$$= -\sum_{j\neq i}\exp\left(\beta \boldsymbol{X}_{[i,:]}^\top \boldsymbol{A}\boldsymbol{X}_{[j,:]}\right)\boldsymbol{A}^\top \boldsymbol{X}_{[j,:]} - \sum_{j\neq i}\exp\left(\beta \boldsymbol{X}_{[i,:]}^\top \boldsymbol{A}\boldsymbol{X}_{[j,:]}\right)\boldsymbol{A}\boldsymbol{X}_{[j,:]} \tag{33}$$

$$- \exp\left(\beta \boldsymbol{X}_{[i,:]}^\top \boldsymbol{A}\boldsymbol{X}_{[i,:]}\right)(\boldsymbol{A}^\top + \boldsymbol{A})\boldsymbol{X}_{[i,:]} \tag{34}$$

$$= -\sum_{j}\exp\left(\beta \boldsymbol{X}_{[i,:]}^\top \boldsymbol{A}\boldsymbol{X}_{[j,:]}\right)(\boldsymbol{A}^\top + \boldsymbol{A})\boldsymbol{X}_{[j,:]}. \tag{35}$$

Under the given condition on the weights, $\boldsymbol{W}^V = (\boldsymbol{A}^\top + \boldsymbol{A})/2$, we have:

$$\Delta_{[i,:]} = (\mathrm{softmax}(\beta \boldsymbol{X}\boldsymbol{A}\boldsymbol{X}^\top)\boldsymbol{X}(\boldsymbol{A}^\top + \boldsymbol{A}))_{[i,:]}^\top/2 \tag{36}$$

$$= \sum_{j}\frac{\exp\left(\beta \boldsymbol{X}_{[i,:]}^\top \boldsymbol{A}\boldsymbol{X}_{[j,:]}\right)}{Z_i}(\boldsymbol{A}^\top + \boldsymbol{A})\boldsymbol{X}_{[j,:]}/2, \tag{37}$$

where $Z_i = \sum_{j'}\exp\left(\beta \boldsymbol{X}_{[i,:]}^\top \boldsymbol{A}\boldsymbol{X}_{[j',:]}\right)$ is the normalization term.

Then, we have,

$$\frac{dE_{\mathrm{single}}(\boldsymbol{X})}{dt} = \frac{dE_{\mathrm{single}}(\boldsymbol{X})}{d\boldsymbol{X}}\cdot \frac{d\boldsymbol{X}}{dt} \tag{38}$$

$$= -\sum_{i}\left(\frac{dE_{\mathrm{single}}(\boldsymbol{X})}{d\boldsymbol{X}_{[i,:]}}\cdot (\boldsymbol{I}_D - \boldsymbol{X}_{[i,:]}\boldsymbol{X}_{[i,:]}^\top)\Delta_{[i,:]}\right) \tag{39}$$

$$= -\sum_{i}\left(\sum_{j}\exp\left(\beta \boldsymbol{X}_{[i,:]}^\top \boldsymbol{A}\boldsymbol{X}_{[j,:]}\right)(\boldsymbol{A}^\top + \boldsymbol{A})\boldsymbol{X}_{[j,:]}\cdot (\boldsymbol{I}_D - \boldsymbol{X}_{[i,:]}\boldsymbol{X}_{[i,:]}^\top)\Delta_{[i,:]}\right) \tag{40}$$

$$= -2\sum_{i}\left(Z_i\Delta_{[i,:]}\cdot (\boldsymbol{I}_D - \boldsymbol{X}_{[i,:]}\boldsymbol{X}_{[i,:]}^\top)\Delta_{[i,:]}\right) \tag{41}$$

$$= -2\sum_{i}\left(Z_i\Delta_{[i,:]}^\top(\boldsymbol{I}_D - \boldsymbol{X}_{[i,:]}\boldsymbol{X}_{[i,:]}^\top)\Delta_{[i,:]}\right) \tag{42}$$

$$\leq 0, \tag{43}$$

where, in the last inequality, we used the fact that the matrix $\boldsymbol{I}_D - \boldsymbol{X}_{[i,:]}\boldsymbol{X}_{[i,:]}^\top$ is positive semi-definite. $\square$

### A.3 Proof of Proposition 4.2

**Proposition 4.2 is restated.** *Consider the continuous-time dynamics for multi-head SA without projection: $d\boldsymbol{X}/dt = \sum_{h=1}^H SA_h(\boldsymbol{X})$. An energy function*

$$E_{multi}(\boldsymbol{X}) = -\sum_{h}\sum_{i,j}\exp\left(\beta \boldsymbol{X}_{[i,:]}^\top \boldsymbol{W}_h^Q \boldsymbol{W}_h^{K\top}\boldsymbol{X}_{[j,:]}\right) \tag{44}$$

*is monotonically decreasing as $dE_{multi}(\boldsymbol{X})/dt \leq 0$ under the condition*

$$\boldsymbol{W}_h^V = (\boldsymbol{W}_h^K\boldsymbol{W}_h^{Q\top} + \boldsymbol{W}_h^Q\boldsymbol{W}_h^{K\top})/2, \quad \boldsymbol{W}_h^Q\boldsymbol{W}_h^{K\top} = \boldsymbol{U}_{1,h}\boldsymbol{U}_{2,h}^\top, \tag{45}$$

*where $\boldsymbol{U}_{1(2),h} \in \mathbb{R}^{D\times D/(2H)}$ ($h \in [1,H]$) satisfies the orthogonality condition $\boldsymbol{U}_{k,h}^\top\boldsymbol{U}_{k',h'} = \delta_{hh'}\delta_{kk'}\boldsymbol{I}_{D/(2H)}$.*

*Proof.* Let $\Delta_h = \mathrm{softmax}(\beta \boldsymbol{X}\boldsymbol{W}_h^Q\boldsymbol{W}_h^{K\top}\boldsymbol{X}^\top)\boldsymbol{X}\boldsymbol{W}_h^V)$ and let $\boldsymbol{A}_h = \boldsymbol{W}_h^Q\boldsymbol{W}_h^{K\top}$. The first-order derivative of $E_{\mathrm{multi}}(\boldsymbol{X})$ with respect to $\boldsymbol{X}_{[i,:]}$ is, similarly to the single-head case, given by:

$$\frac{\partial E_{\mathrm{multi}}(\boldsymbol{X})}{\partial \boldsymbol{X}_{[i,:]}} = -\sum_{h}\sum_{j}\exp\left(\beta \boldsymbol{X}_{[i,:]}^\top \boldsymbol{A}_h\boldsymbol{X}_{[j,:]}\right)(\boldsymbol{A}_h^\top + \boldsymbol{A}_h)\boldsymbol{X}_{[j,:]} \tag{46}$$

Under the given condition on the weights, similar to the single-head case, we have:

$$\Delta_{h[i,:]} = \sum_j \frac{\exp\left(\beta \boldsymbol{X}_{[i,:]}^\top \boldsymbol{A}_h \boldsymbol{X}_{[j,:]}\right)}{Z_{h,i}} (\boldsymbol{A}_h^\top + \boldsymbol{A}_h) \boldsymbol{X}_{[j,:]}/2 \tag{47}$$

where $Z_{h,i} = \sum_{j'} \exp\left(\beta \boldsymbol{X}_{[i,:]}^\top \boldsymbol{A}_h \boldsymbol{X}_{[j',:]}\right)$ is the normalization term. Then, we have,

$$\frac{dE_{\text{multi}}(\boldsymbol{X})}{dt} = \frac{dE_{\text{multi}}(\boldsymbol{X})}{d\boldsymbol{X}} \cdot \frac{d\boldsymbol{X}}{dt} \tag{48}$$

$$= -\sum_i \left( \frac{\partial E_{\text{multi}}(\boldsymbol{X})}{\partial \boldsymbol{X}_{[i,:]}} \cdot \sum_h \Delta_{h[i,:]} \right) \tag{49}$$

$$= -\sum_i \left( \sum_h \sum_j \exp\left(\beta \boldsymbol{X}_{[i,:]}^\top \boldsymbol{A}_h \boldsymbol{X}_{[j,:]}\right) (\boldsymbol{A}_h^\top + \boldsymbol{A}_h) \boldsymbol{X}_{[j,:]} \cdot \sum_h \Delta_{h[i,:]} \right) \tag{50}$$

$$= -2\sum_i \left( \sum_h Z_{h,i} \Delta_{h[i,:]} \cdot \sum_h \Delta_{h[i,:]} \right) \tag{51}$$

$$= -2\sum_i \sum_h Z_{h,i} \|\Delta_{h[i,:]}\|^2 \tag{52}$$

$$\leq 0, \tag{53}$$

where we use the fact that for $h \neq h'$,

$$\boldsymbol{A}_h^\top \boldsymbol{A}_{h'} = \boldsymbol{U}_{2,h} \boldsymbol{U}_{1,h}^\top \boldsymbol{U}_{1,h'} \boldsymbol{U}_{2,h'}^\top = \boldsymbol{O}, \quad \boldsymbol{A}_h \boldsymbol{A}_h^\top = \boldsymbol{O}, \tag{54}$$

$$\boldsymbol{A}_h \boldsymbol{A}_{h'} = \boldsymbol{U}_{1,h} \boldsymbol{U}_{2,h}^\top \boldsymbol{U}_{1,h'} \boldsymbol{U}_{2,h'}^\top = \boldsymbol{O}, \tag{55}$$

and thus

$$\Delta_{h[i,:]} \cdot \Delta_{h'[i,:]} = \Delta_{h[i,:]}^\top \Delta_{h'[i,:]} \tag{56}$$

$$= \left( \sum_j \frac{\exp\left(\beta \boldsymbol{X}_{[i,:]}^\top \boldsymbol{A}_h \boldsymbol{X}_{[j,:]}\right)}{Z_{h,i}} \boldsymbol{X}_{[j,:]} \right)^\top \tag{57}$$

$$(\boldsymbol{A}_h^\top + \boldsymbol{A}_h)(\boldsymbol{A}_{h'} + \boldsymbol{A}_{h'}^\top) \sum_j \frac{\exp\left(\beta \boldsymbol{X}_{[i,:]}^\top \boldsymbol{A}_{h'} \boldsymbol{X}_{[j,:]}\right)}{Z_{h',i}} \boldsymbol{X}_{[j,:]}/4 \tag{58}$$

$$= 0. \tag{59}$$

$\square$

## A.4  Proof of Proposition 5.1

**Proposition 5.1 is restated.** *Suppose that, in the update of ItrSA* (9)*, the input to the normalization layer satisfies $\|\boldsymbol{X}_{[i,:]} + \eta \Delta \boldsymbol{X}_{[i,:]}\| \geq R$ for all $i \in [1, S]$. Then, the spectral norm of the Jacobian satisfies the upper bound*

$$\left\| \frac{\partial \operatorname{RMSNorm}(\boldsymbol{X} + \eta \Delta \boldsymbol{X})}{\partial \boldsymbol{X}} \right\|_2 \leq \frac{\max_j(|\gamma_j|)}{R} \left(1 + \eta \|\boldsymbol{J}_{MSA}(\boldsymbol{X})\|_2\right), \tag{60}$$

*where $\boldsymbol{J}_{MSA}(\boldsymbol{X}) := \partial \operatorname{MSA}(\boldsymbol{X})/\partial \boldsymbol{X}$ denotes the Jacobian of MSA.*

*Proof.* First, for any vector $a \in \mathbb{R}^D$, the eigenvalues of the matrix $I_D - \frac{aa^\top}{\|a\|^2}$ are 1 (with multiplicity $D - 1$) and 0 (with multiplicity 1). Hence,

$$\left\| I_D - \frac{aa^\top}{\|a\|^2} \right\|_2 = 1. \tag{61}$$

Using Lemma A.3, we have

$$\left\| \frac{\partial \operatorname{RMSNorm}(\boldsymbol{Y})}{\partial \boldsymbol{Y}} \right\|_2 = \left\| \operatorname{blockdiag}\left( \left\{ \frac{1}{\|\boldsymbol{Y}_{[i,:]}\|} \operatorname{diag}(\boldsymbol{\gamma}) \left( I_D - \frac{\boldsymbol{Y}_{[i,:]}\boldsymbol{Y}_{[i,:]}^\top}{\|\boldsymbol{Y}_{[i,:]}\|^2} \right) \right\}_i \right) \right\|_2 \tag{62}$$

$$= \max_i \left\| \frac{1}{\|\boldsymbol{Y}_{[i,:]}\|} \operatorname{diag}(\boldsymbol{\gamma}) \left( I_D - \frac{\boldsymbol{Y}_{[i,:]}\boldsymbol{Y}_{[i,:]}^\top}{\|\boldsymbol{Y}_{[i,:]}\|^2} \right) \right\|_2 \tag{63}$$

$$\leq \frac{\max_j |\gamma_j|}{R}. \tag{64}$$

Therefore, setting $\boldsymbol{Y} = \boldsymbol{X} + \eta\Delta\boldsymbol{X} = \boldsymbol{X} + \eta\left(\boldsymbol{C} + \operatorname{MSA}(\boldsymbol{X})\right)$, we have

$$\left\| \frac{\partial \operatorname{RMSNorm}(\boldsymbol{X} + \eta\Delta\boldsymbol{X})}{\partial \boldsymbol{X}} \right\|_2 = \left\| \frac{\partial \operatorname{RMSNorm}(\boldsymbol{Y})}{\partial \boldsymbol{Y}} \frac{\partial \boldsymbol{Y}}{\partial \boldsymbol{X}} \right\|_2 \tag{65}$$

$$\leq \left\| \frac{\partial \operatorname{RMSNorm}(\boldsymbol{Y})}{\partial \boldsymbol{Y}} \right\|_2 \left\| \frac{\partial \boldsymbol{Y}}{\partial \boldsymbol{X}} \right\|_2 \tag{66}$$

$$\leq \frac{\max_j |\gamma_j|}{R} \|\boldsymbol{I}_{SD} + \eta\boldsymbol{J}_{\operatorname{MSA}}\|_2 \tag{67}$$

$$\leq \frac{\max_j (|\gamma_j|)}{R} \left( 1 + \eta\|\boldsymbol{J}_{\operatorname{MSA}}(\boldsymbol{X})\|_2 \right). \tag{68}$$

$\square$

Here, we can also show that, in the limit $\eta \to \infty$, the Jacobian norm remains $O(1)$. Define, for each $i$, the projection

$$\boldsymbol{P}_i := \boldsymbol{I}_D - \frac{\boldsymbol{Y}_{[i,:]}\boldsymbol{Y}_{[i,:]}^\top}{\|\boldsymbol{Y}_{[i,:]}\|_2^2}, \tag{69}$$

so that $\|\boldsymbol{P}_i\|_2 = 1$. Let $\boldsymbol{D} := \operatorname{diag}(\boldsymbol{\gamma})$, and define the block factors

$$\boldsymbol{A}_i(\eta) := \frac{1}{\|\boldsymbol{X}_{[i,:]} + \eta\,\Delta\boldsymbol{X}_{[i,:]}\|_2} \boldsymbol{D}\,\boldsymbol{P}_i, \qquad \boldsymbol{A}(\eta) := \operatorname{blockdiag}\left( \{\boldsymbol{A}_i(\eta)\}_{i=1}^N \right). \tag{70}$$

By the triangle inequality, for each $i$,

$$\|\boldsymbol{X}_{[i,:]} + \eta\,\Delta\boldsymbol{X}_{[i,:]}\|_2 \geq \eta\,\|\Delta\boldsymbol{X}_{[i,:]}\|_2 - \|\boldsymbol{X}_{[i,:]}\|_2. \tag{71}$$

Recall that $\boldsymbol{X}$ is the output of the previous layer, so that

$$\boldsymbol{X} = \operatorname{RMSNorm}(\boldsymbol{Z}) \tag{72}$$

for some $\boldsymbol{Z} \in \mathbb{R}^{S \times D}$. Therefore,

$$\|\boldsymbol{X}_{[i,:]}\|_2 = \|\operatorname{RMSNorm}(\boldsymbol{Z})_{[i,:]}\|_2 \tag{73}$$

$$= \left\| \operatorname{diag}(\boldsymbol{\gamma}) \frac{\boldsymbol{Z}_{[i,:]}}{\|\boldsymbol{Z}_{[i,:]}\|_2} \right\|_2 \tag{74}$$

$$= \frac{\sqrt{\sum_{j=1}^D \gamma_j^2 Z_{[i,j]}^2}}{\|\boldsymbol{Z}_{[i,:]}\|_2} \tag{75}$$

$$\leq \max_j |\gamma_j| \frac{\sqrt{\sum_{j=1}^D Z_{[i,j]}^2}}{\|\boldsymbol{Z}_{[i,:]}\|_2} \tag{76}$$

$$= \max_j |\gamma_j| \tag{77}$$

for each $i = 1, \ldots, S$.

Assume that $\min_i \|\Delta\boldsymbol{X}_{[i,:]}\|_2 \geq \varepsilon$ for some constant $\varepsilon > 0$. Then, for a sufficiently large $\eta$ satisfying $\eta \geq \max_j |\gamma_j|/\varepsilon$, we have

$$\eta \geq 2 \frac{\|\boldsymbol{X}_{[i,:]}\|_2}{\|\Delta\boldsymbol{X}_{[i,:]}\|_2}. \tag{78}$$

Hence,

$$\|\boldsymbol{X}_{[i,:]} + \eta\,\Delta\boldsymbol{X}_{[i,:]}\|_2 \geq \eta\,\|\Delta\boldsymbol{X}_{[i,:]}\|_2 - \|\boldsymbol{X}_{[i,:]}\|_2 \tag{79}$$

$$\geq \tfrac{\eta}{2}\,\|\Delta\boldsymbol{X}_{[i,:]}\|_2. \tag{80}$$

Using submultiplicativity and $\|\boldsymbol{P}_i\|_2 = 1$, we obtain

$$\|\boldsymbol{A}_i(\eta)\|_2 = \frac{\|\boldsymbol{D}\,\boldsymbol{P}_i\|_2}{\|\boldsymbol{X}_{[i,:]} + \eta\,\Delta\boldsymbol{X}_{[i,:]}\|_2} \tag{81}$$

$$\leq \frac{\|\boldsymbol{D}\|_2}{\|\boldsymbol{X}_{[i,:]} + \eta\,\Delta\boldsymbol{X}_{[i,:]}\|_2} \tag{82}$$

$$\leq \frac{2\,\|\boldsymbol{D}\|_2}{\eta\,\|\Delta\boldsymbol{X}_{[i,:]}\|_2}, \tag{83}$$

for all $i = 1, \ldots, S$.

For a block-diagonal matrix, the operator norm equals the maximum block norm; thus

$$\|\boldsymbol{A}(\eta)\|_2 = \max_i \|\boldsymbol{A}_i(\eta)\|_2 \tag{84}$$

$$\leq \max_i \frac{2\,\|\boldsymbol{D}\|_2}{\eta\,\|\Delta\boldsymbol{X}_{[i,:]}\|_2} \tag{85}$$

$$= \frac{2\,\|\boldsymbol{D}\|_2}{\eta\,\min_i \|\Delta\boldsymbol{X}_{[i,:]}\|_2} \tag{86}$$

$$\leq \frac{2\,\|\boldsymbol{D}\|_2}{\eta\,\varepsilon}. \tag{87}$$

Therefore, we have

$$\|\boldsymbol{A}(\eta)\,(\boldsymbol{I}_{ND} + \eta\,\boldsymbol{J}_{\mathrm{MSA}})\|_2 \leq \|\boldsymbol{A}(\eta)\|_2\,(\|\boldsymbol{I}_{ND}\|_2 + \eta\,\|\boldsymbol{J}_{\mathrm{MSA}}\|_2) \tag{88}$$

$$\leq \frac{2\,\|\boldsymbol{D}\|_2}{\varepsilon}\left(\|\boldsymbol{J}_{\mathrm{MSA}}\|_2 + \frac{1}{\eta}\right). \tag{89}$$

Since Eq. (16) makes a constant upper-bound of $\|\boldsymbol{J}_{\mathrm{MSA}}\|_2$, for $\eta \to \infty$, we obtain

$$\left\|\frac{\partial\,\mathrm{RMSNorm}(\boldsymbol{X} + \eta\,\Delta\boldsymbol{X})}{\partial\boldsymbol{X}}\right\|_2 = O(1). \tag{90}$$

### A.5 Derivation of the eigenvalue bound in oscillatory cases (Section 5.2)

We show that all eigenvalues $\lambda_j$ of the Jacobian

$$\boldsymbol{J}(x) = \left(\boldsymbol{I}_D - \frac{\boldsymbol{y}\boldsymbol{y}^\top}{\|\boldsymbol{y}\|^2}\right)\frac{1}{\|\boldsymbol{y}\|}(\boldsymbol{I}_D + \eta\boldsymbol{\Omega}), \tag{91}$$

satisfy $|\lambda_j| \leq 1$, where $\boldsymbol{y} = (\boldsymbol{I}_D + \eta\boldsymbol{\Omega})\boldsymbol{x}$.

We begin by computing the norm of $\boldsymbol{y}$:

$$\|\boldsymbol{y}\|^2 = \boldsymbol{x}^\top(\boldsymbol{I}_D + \eta\boldsymbol{\Omega}^\top)(\boldsymbol{I}_D + \eta\boldsymbol{\Omega})\boldsymbol{x} \tag{92}$$

$$= \boldsymbol{x}^\top\boldsymbol{x} + \eta\boldsymbol{x}^\top\boldsymbol{\Omega}\boldsymbol{x} + \eta\boldsymbol{x}^\top\boldsymbol{\Omega}^\top\boldsymbol{x} + \eta^2\boldsymbol{x}^\top\boldsymbol{\Omega}^\top\boldsymbol{\Omega}\boldsymbol{x} \tag{93}$$

$$= 1 + \eta^2\|\boldsymbol{\Omega}\boldsymbol{x}\|^2, \tag{94}$$

where we used the fact that for an antisymmetric matrix $\boldsymbol{\Omega}$, $\boldsymbol{x}^\top\boldsymbol{\Omega}\boldsymbol{x} = 0$. Note also that an antisymmetric matrix has eigenvalues of the form $\pm i\omega_j$, where $\omega_j \geq 0$ $(j = 1, 2, \ldots)$. For simplicity, assume all eigenvalues have identical magnitude $\omega_j = \omega$. Then, we have

$$\|\boldsymbol{y}\|^2 = 1 + \eta^2\omega^2. \tag{95}$$

We also use the facts that

$$\left\|\boldsymbol{I}_D - \frac{\boldsymbol{y}\boldsymbol{y}^\top}{\|\boldsymbol{y}\|^2}\right\|_2 \leq 1 \tag{96}$$

and

$$\|\boldsymbol{I}_D + \eta\boldsymbol{\Omega}\|_2 = |1 \pm i\eta\omega| = \sqrt{1 + \eta^2\omega^2}. \tag{97}$$

Combining these, we obtain the following bound on the spectral norm of $\boldsymbol{J}(\boldsymbol{x})$:

$$\|\boldsymbol{J}(\boldsymbol{x})\|_2 = \left\|\left(\boldsymbol{I}_D - \frac{\boldsymbol{y}\boldsymbol{y}^\top}{\|\boldsymbol{y}\|^2}\right)\frac{1}{\|\boldsymbol{y}\|}(\boldsymbol{I}_D + \eta\boldsymbol{\Omega})\right\|_2 \tag{98}$$

$$\leq \frac{1}{\|\boldsymbol{y}\|}\|\boldsymbol{I}_D + \eta\boldsymbol{\Omega}\|_2 \tag{99}$$

$$\leq \frac{\sqrt{1 + \eta^2\omega^2}}{\sqrt{1 + \eta^2\omega^2}} = 1. \tag{100}$$

This implies that all eigenvalues of $\boldsymbol{J}(\boldsymbol{x})$ satisfy $|\lambda_j| \leq 1$.

## B Experimental details

### B.1 Experimental setup

We solved Sudoku task, which is a puzzle played on a $9 \times 9$ grid, where some of the cells are pre-filled with digits from 1 to 9, and the remaining cells are left blank. The objective is to fill in the blank cells such that each 1) row, 2) column, and 3) $3 \times 3$ subgrid contains each digit exactly once.

In our experiments, we used two Sudoku datasets: the SATNet [Wang et al., 2019] and RRN dataset [Palm et al., 2018]. The key differences between the two are that the RRN dataset is more difficult (with only 17–34 given digits compared to 31–42 in SATNet) and larger in size (198k samples vs. 10k samples). Following Miyato et al. [2025], we used the SATNet dataset for training as in-distribution (ID) data and the RRN dataset as out-of-distribution (OOD) data. This setup allows us to evaluate the ability of models to generalize to more challenging settings.

We primarily followed Miyato et al. [2025] and used their official implementation[1], setting the dimension of oscillators of AKOrN to $N = 8$. The readout module of AKOrN (described in Appendix C.2) was also incorporated into our ItrSA model. We used the Adam optimizer [Kingma and Ba, 2015] and trained for 100 epochs with batch size 100. For all settings, we tuned the learning rate over $\{1 \times 10^{-6}, 5 \times 10^{-6}, \ldots, 1 \times 10^{-3}\}$ and, for regularization methods in Figure 5, the parameter $\lambda$ over $\{1 \times 10^{-8}, 1 \times 10^{-7}, \ldots, 1 \times 10^{-1}\}$, selecting values based on OOD accuracy at iteration $T = 16$. All experiments were conducted on NVIDIA H200 GPUs, and we run experiments with 5 different random seeds.

We also conducted experiments on the CIFAR-10 dataset [Krizhevsky et al.]. See table S.1 for training and model configurations.

Table S.1: Training and model configurations.

| Parameter | Sudoku | CIFAR-10 |
|---|---|---|
| Hidden dimension $D$ | 512 | 384 |
| Number of heads $H$ | 8 | 8 |
| Initial value of $\eta$ | 1.0 | 1.0 |
| Batch size | 100 | 128 |
| Number of epochs | 100 | 200 |

### B.2 Single-head generalized symmetric SA

For single-head generalized symmetric SA, we define

$$R_{\text{E-single}} := \left\|\boldsymbol{W}_1^V\boldsymbol{W}_1^O - (\boldsymbol{W}_1^V\boldsymbol{W}_1^O)^\top\right\|_F^2, \tag{101}$$

under the condition that $H = 1$. If $R_{\text{E-single}} = 0$, setting $\boldsymbol{W}^V = \boldsymbol{W}_1^V\boldsymbol{W}_1^O$ satisfies the condition on $\boldsymbol{W}^V$ described in Proposition 4.1.

---

[1]https://github.com/autonomousvision/akorn

Table S.2: Regularization coefficients used in Figure 5.

| Method | Value |
|--------|-------|
| E-single | 1e-6 |
| E-multi | 1e-4 |
| Spec (ItrSA) | 1e-4 |
| Spec (AKOrN) | 1e-5 |

## B.3 Lyapunov exponent

The Lyapunov exponent quantifies the exponential rate at which nearby trajectories in a dynamical system diverge. For a discrete-time system $\boldsymbol{x}^{(t+1)} = \boldsymbol{f}(\boldsymbol{x}^{(t)})$, the Lyapunov spectrum $\{\lambda_i\}$ is defined as:

$$\lambda_i = \lim_{T \to \infty} \frac{1}{2T} \log |\alpha_i^{(T)}|, \tag{102}$$

where $\alpha_i^{(T)}$ is the $i$-th eigenvalue of the positive semi-definite matrix

$$\Lambda^{(T)} = \left( \frac{d\boldsymbol{f}^T(\boldsymbol{x}^{(0)})}{d\boldsymbol{x}^{(0)}} \right)^\top \frac{d\boldsymbol{f}^T(\boldsymbol{x}^{(0)})}{d\boldsymbol{x}^{(0)}}, \tag{103}$$

and $\boldsymbol{f}^T$ denotes the $T$-fold composition of the function $\boldsymbol{f}$. The *maximum Lyapunov exponent* is then defined as

$$\lambda_{\max} := \max_i \lambda_i. \tag{104}$$

To mitigate numerical sensitivity, we also use *mean Lyapunov exponent*

$$\lambda_{\mean} := \frac{1}{M} \sum_{i=1}^{M} \lambda_i, \tag{105}$$

where $M$ denotes the number of Lyapunov exponents.

In our experiments, we approximated the Lyapunov spectrum using a finite time horizon of $T = 16$ on a randomly selected sample. For models without normalization and symmetric SA, we trained them for only one epoch, as full training was not feasible due to instability. To evaluate how the Lyapunov exponent varies, we adjusted the input scaling of $\boldsymbol{X}$, the step size $\eta$, and the norms of the value projection weights, $\|\boldsymbol{W}_h^V\|$ and $\|\boldsymbol{W}_h^O\|$, in the SA update of $\boldsymbol{X}$.

## B.4 Details of other figures

For Figure 2c, we computed the eigenvalues of the Jacobian matrix at $T = 16$ on a randomly selected sample from the Sudoku dataset. For the model with normalization, we used the fully trained model. For models without normalization, we followed the same setup as in the Lyapunov experiments and used models trained for only one epoch.

For the computation of the SA's Jacobian in Figure 1, we used the CCDV arXiv summarization dataset [Cohan et al., 2018], as it provides text data suitable for varying the number of tokens. We used an initialized SA and computed the Jacobian and SA followed by normalization over 500 randomly selected samples. The norm of tokens was set to $R = 100$ and their dimensions to $D = 256$.

## B.5 Additional results

### B.5.1 Effects of model structure and regularization

**Number of attention heads.** To further examine the effect of normalization, we computed Lyapunov exponents while varying the number of attention heads $H$. Because the maximum Lyapunov exponent is numerically sensitive, we report the *mean* Lyapunov exponent instead. The results in Figure S.1 show that models with few heads ($H \in \{1, 2\}$) perform poorly, whereas $H = 8$ achieve the highest accuracy. Moreover, the mean Lyapunov exponent is positively correlated with accuracy, suggesting that more dynamic states are associated with better performance.

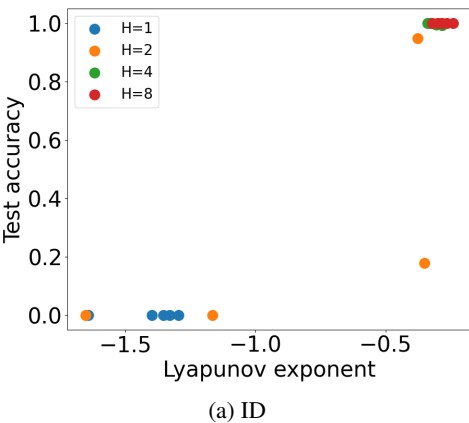
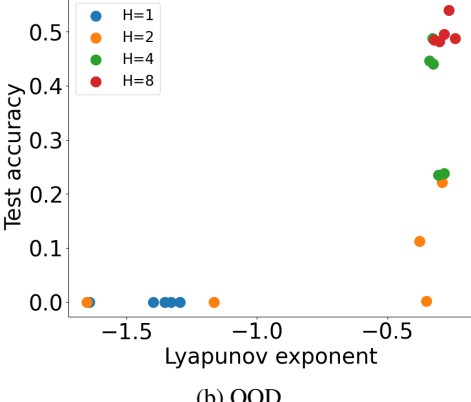

(a) ID                                          (b) OOD

Figure S.1: Effect of the number of attention heads $H$ on mean Lyapunov exponent and accuracy in the Sudoku dataset.

$\gamma$ **in** $\mathrm{RMSNorm}$. Table S.3 presents the values of the $\gamma$ parameter learned by ItrSA. The results indicate that the trained models exhibit small $\gamma$ values, with $\max_j |\gamma_j| < 1$.

Table S.3: $\gamma$ in $\mathrm{RMSNorm}$ with different $N$.

| $N$ | $\|\gamma\|$ | $\max_j |\gamma_j|$ |
|---|---|---|
| 4 | $0.229 \pm 0.000$ | $0.229 \pm 0.000$ |
| 8 | $0.031 \pm 0.002$ | $0.052 \pm 0.000$ |
| 16 | $0.098 \pm 0.006$ | $0.098 \pm 0.006$ |
| 32 | $0.348 \pm 0.000$ | $0.348 \pm 0.000$ |
| 64 | $0.489 \pm 0.001$ | $0.489 \pm 0.001$ |
| 128 | $0.841 \pm 0.000$ | $0.841 \pm 0.000$ |
| 256 | $0.738 \pm 0.000$ | $0.738 \pm 0.000$ |
| 512 | $0.811 \pm 0.000$ | $0.811 \pm 0.000$ |

**Further results on regularization.** In Figure S.2, we evaluate the effect of the proposed multi-head energy (Proposition 4.2) on both the mean Lyapunov exponent and accuracy. As the regularization strength $\lambda$ increases, both metrics consistently decrease. Even at $\lambda = 0$, accuracy is lower than the original model due to the orthogonality constraint. The "Hard" constraint yields the lowest accuracy and the smallest Lyapunov exponent. Overall, these results indicate that stronger regularization suppresses the Lyapunov exponent but also degrades accuracy, suggesting that while multi-head energy regularization encourages more convergent dynamics, it does not necessarily yield better performance.

In Figure S.3 we plot the case where we used $N = 8$ as the oscillator dimension of AKOrN. AKOrN achieves the best performance and spectral regularization is also effective.

### B.5.2   Distribution of Lyapunov exponents

Figure S.4 shows the distibution of the Lyapunov exponent.

### B.5.3   Lyapunov exponents in language modeling tasks

To evaluate in a more realistic scenario, we conducted language modeling experiments on the BabyLM Challenge dataset (2023, 10M version) [Diehl Martinez et al., 2023], comparing our ItrSA model with GPT-2 [Radford et al., 2019]. We trained for 30 epochs using the AdamW optimizer. As shown in Table S.4, the maximum Lyapunov exponent (MLE) for ItrSA is slightly positive, which is consistent with our other tasks and suggests mildly chaotic dynamics. We also show the loss values in the case of GPT-2 as a reference which confirms that the performance of our ItrSA can become comparable to that of GPT-2.

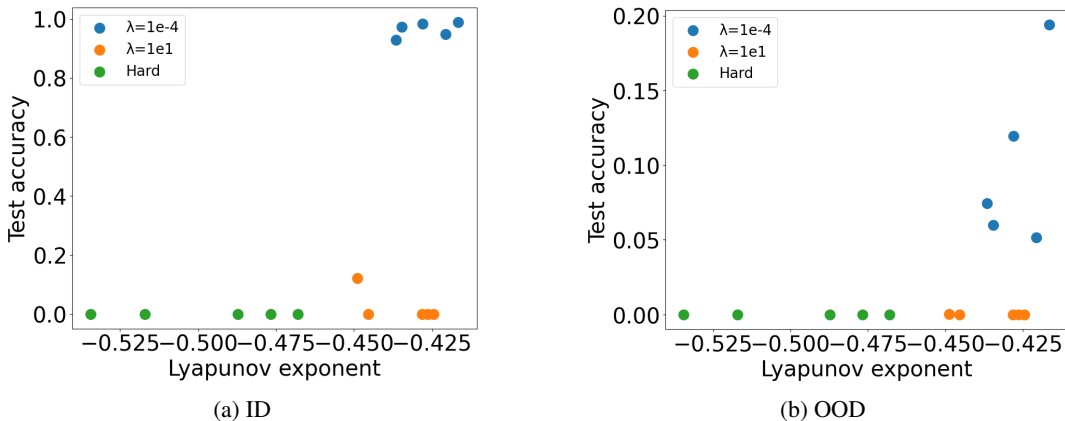

|          |          |
|:--------:|:--------:|
| (a) ID   | (b) OOD  |

Figure S.2: Effect of multi-head energy on the mean Lyapunov exponent and accuracy for the Sudoku dataset. $\lambda$ denotes the regularization coefficient of Eq. (20), and "Hard" indicates the hard constraint in Proposition 4.2.

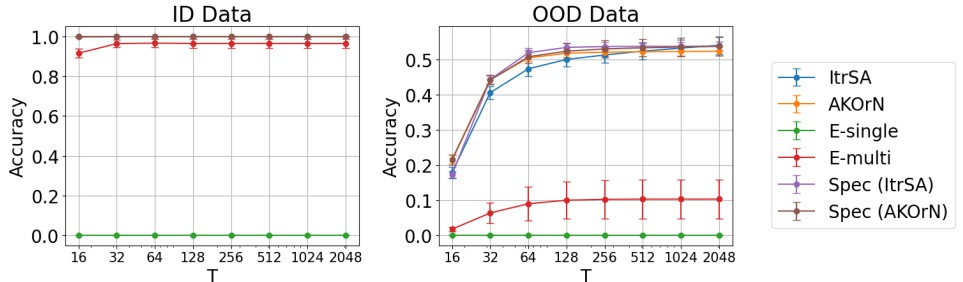

Figure S.3: Energy-based regularization ("E-single" and "E-multi") underperforms the original methods, while Jacobian spectral regularization ("Spec") outperforms. We used $N = 8$ for AKOrN.

### B.5.4 Sensitivity to initial conditions

In our study, we define "criticality" as the point at which the largest Lyapunov exponent takes zero. In our experiments, when the maximum Lyapunov exponent gets slightly positive, we observed behaviors consistent with widely accepted notions of chaos, particularly sensitivity to initial conditions as follows.

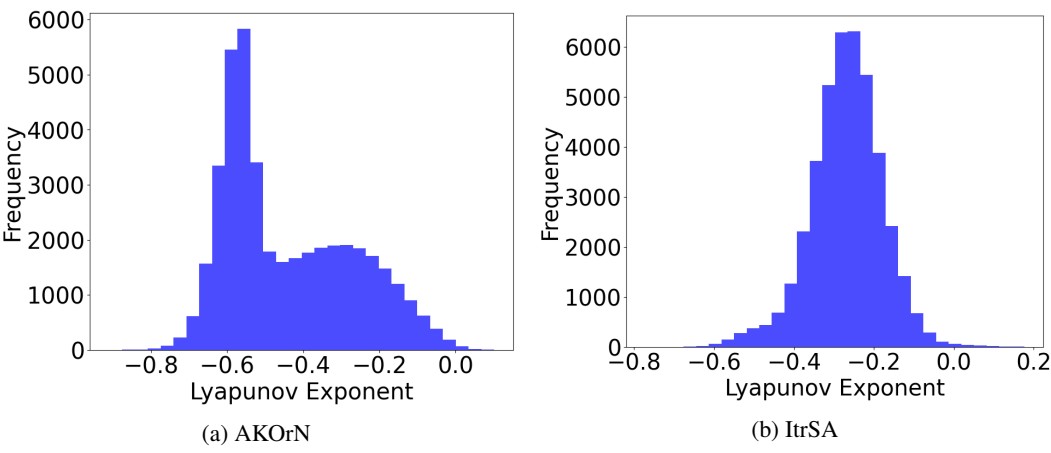

|           |           |
|:---------:|:---------:|
| (a) AKOrN | (b) ItrSA |

Figure S.4: Distribution of the Lyapunov exponent.

Table S.4: Language modeling results. MLE denotes the maximum Lyapunov exponent.

| Model | # Parameters | Training loss | Validation loss | MLE |
|---|---|---|---|---|
| ItrSA | 96.2M | 4.31 | 5.67 | $0.196 \pm 0.016$ |
| GPT-2 | 124M | 1.49 | 5.58 | - |

Starting from an input, we ran the loop of the ItrSA model for 128 steps to obtain $x_{t=0}$. We then repeated the run from a perturbed input $x_{t=0} + \epsilon$, where $\epsilon \sim N(0, 10^{-3})$ is added at $t = 0$. From each trajectory, we sampled 300 equally spaced points between $t = 0$ and $t = 10000$. The results are shown in Figure S.5.

Figure S.5a plots the L1 distance between the two trajectories. In Figures S.5b and S.5c, we visualize a single coordinate of five trajectories. Figure S.5b shows the complete trajectories. Since we observed that the trajectories rapidly oscillate between two clusters and the visualization was subtle, Figure S.5c provides a zoomed-in view of one cluster (values $> 0.1$).

Overall, the distance between the original and perturbed trajectories increases exponentially over time, demonstrating chaotic behavior in the system.

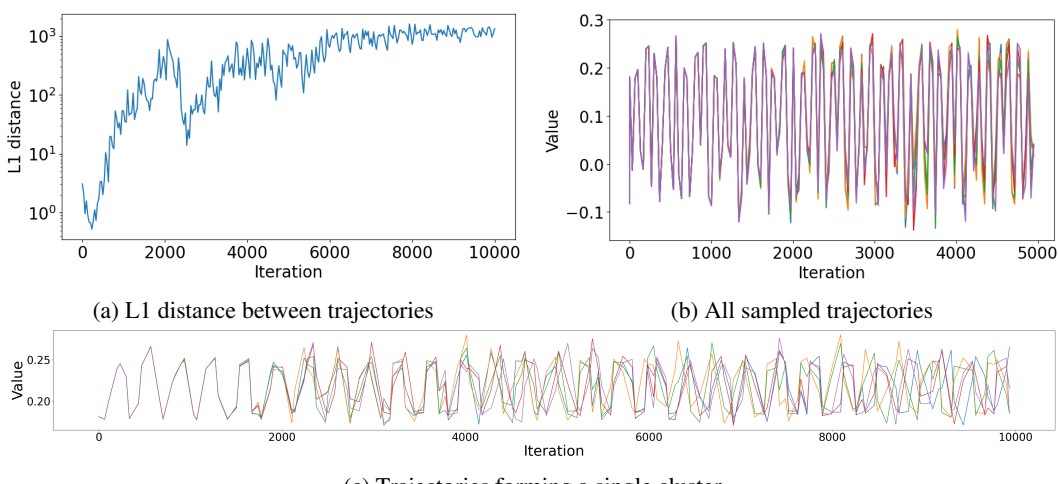

(a) L1 distance between trajectories

(b) All sampled trajectories

(c) Trajectories forming a single cluster

Figure S.5: Sensitivity to initial conditions investigated using the ItrSA model.

#### B.5.5 Jacobian-based interpretation of pseudo-energy

We provide additional insights into the interpretation of pseudo-energy via the Jacobian discussed in Section 6.3. Using Lemma A.3 from Noci et al. [2022], the Jacobian of SA is expressed as

$$
\boldsymbol{J} = \frac{\partial \, \mathrm{SA}(\boldsymbol{X})}{\partial \boldsymbol{X}} = (\boldsymbol{I}_S \otimes \boldsymbol{W}^{V\top} \boldsymbol{X}^\top) \frac{\partial \boldsymbol{P}}{\partial \boldsymbol{X}} + \boldsymbol{P} \otimes \boldsymbol{W}^{V\top}, \tag{106}
$$

where $\boldsymbol{P} := \mathrm{softmax}\left(\boldsymbol{X}\boldsymbol{W}^Q\boldsymbol{W}^{K\top}\boldsymbol{X}^\top/\sqrt{D_H}\right)$.

As our experimental result indicated, we observed that $\boldsymbol{J}\boldsymbol{x}$ aligns well with $\mathrm{vec}(\mathrm{MSA}(\boldsymbol{X}))$. Since each head can be expressed as $\mathrm{vec}(\mathrm{SA}(\boldsymbol{X})) = (\boldsymbol{P} \otimes \boldsymbol{W}^{V\top})\boldsymbol{x}$, our observation implies that in the Jacobian (106), the last term is dominant, that is,

$$
\boldsymbol{J}_t \approx \sum_{h=1}^{H} \boldsymbol{P}_h^{(t)} \otimes \boldsymbol{W}_h^{V\top}. \tag{107}
$$

In other words, the derivative of the attention matrix $\boldsymbol{P}$ is small while that of the value matrix remains significant. In addition, if the derivative of the attention matrix $\boldsymbol{P}$ is sufficiently small over the whole time, the Lipschitz continuity implies that $\boldsymbol{P}$ remains close to its initialization, suggesting a

time-independent Jacobian approximation:

$$J_t \approx \sum_{h=1}^{H} P_h^{(0)} \otimes W_h^{V\top}. \tag{108}$$

Then, the pseudo-energy is approximated by the following quadratic form:

$$E_{\text{pseudo}} \approx -x_t^{\top} \left( \sum_{h=1}^{H} P_h^{(0)} \otimes W_h^{V\top} \right) x_t. \tag{109}$$

Figure S.6 empirically confirmed that both the contribution index and the pseudo-energy behave similarly even under these approximations. PV indicates the contribution index using the approximation (107), and PV ($T = 0$) uses (108). The figure shows that the contribution index can be effectively explained using only the attention matrix at the initialization of inference. The blue curve shows the approximated pseudo-energy (109) and works similarly to the original one. Thus, our Jacobian-based analysis    interprets the pseudo-energy as quantifying exploration in eigenspaces corresponding to large eigenvalues of the attention matrix determined by the initial inference states.

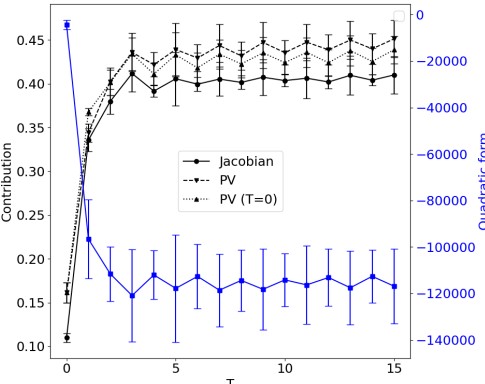

Figure S.6: Contribution and the quandratic form.

### B.5.6    CIFAR-10

For the experiments on the CIFAR-10 dataset [Krizhevsky et al.], we used the same architecture and setup as in the Sudoku experiments. We used the Adam optimizer and tuned the learning rate across $\{1 \times 10^{-6}, 5 \times 10^{-6}, \ldots, 1 \times 10^{-3}\}$ based on the test accuracy at the iteration $T = 16$. We trained models for 200 epochs and set the batch size 128. We used $N = 4$ as the dimension of oscillators of AKOrN.

Figure S.7 shows the Lyapunov exponent on the CIFAR-10 dataset. This result is in the same trend with that on the Sudoku dataset.

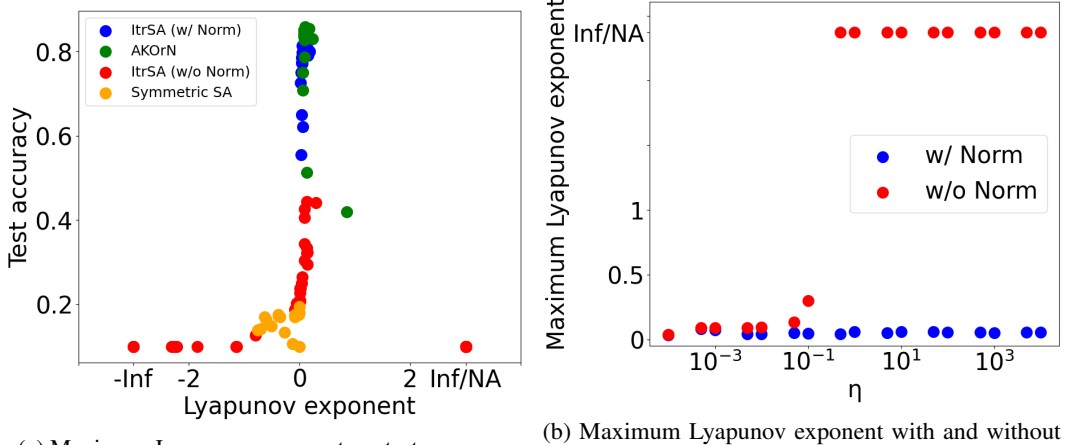

(a) Maximum Lyapunov exponent vs. test accuracy

(b) Maximum Lyapunov exponent with and without normalization

Figure S.7: Lyapunov exponent on the CIFAR-10 dataset.

## C  Other details

### C.1  Extended related work

**Energy-based understanding.**  The Transformer architecture has been a focus of efforts to provide theoretical grounding. Geshkovski et al. [2025, 2023, 2024] formulated recurrent SA dynamics as interactions among tokens ("particles"), enabling theoretical analysis of phenomena such as meta-stable clustering and rank-one collapse. Their continuous-time dynamics monotonically decrease an energy (Lyapunov) function given by a summation over exponential functions, commonly requiring constraints such as single-head attention, hyperspherical token states, and symmetric weights. Karagodin et al. [2024] extended this framework to the case of causal attention masking. Bruno et al. [2025] succeeded in mathematically characterizing the meta-stable clustering as a Wasserstein gradient flow of mean-field token dynamics, with the energy serving as its potential function, although they replaced the softmax function with an unnormalized exponential function and restricted their analysis to identity weight matrices. Yang et al. [2022] considered an exponential energy function similar to that of Geshkovski et al. [2025], describing the Transformer as performing alternating majorization-minimization updates on distinct energy functions. Their approach also accommodates discrete state updates and MLP layers, although it entails complex conditions, including constraints on step sizes and proximity to fixed points. Ramsauer et al. [2021] formalized the cross-attention mechanism as modern Hopfield networks, Hoover et al. [2023], Hu et al. [2025] further developed energy functions for Transformers including self-attentions. We do not address approaches based on Hopfield networks in this work, as they require architectural modifications, such as adding auxiliary signal paths that are absent in standard Transformers, which are beyond our scope.

**Jacobian-based analysis.**  The Jacobian of state updates is fundamental for characterizing neural network dynamics. For example, it has been used to analyze edge-of-chaos behavior for stable signal propagation and gradient control [Boedecker et al., 2012, Poole et al., 2016, Pennington et al., 2017]. Haber and Ruthotto [2017] interpreted forward propagation in neural networks as continuous-time dynamical systems and analyzed their Jacobians to prevent exploding and vanishing gradients. Chang et al. [2019] extended the ODE-based perspective to recurrent neural networks and proposed using anti-symmetric weight matrices to satisfy discrete-time stability conditions. Several studies have explored Jacobian-based regularization techniques. Yoshida and Miyato [2017] proposed spectral norm regularization to reduce sensitivity to input perturbations and improve generalization. Miyato et al. [2018] applied spectral normalization to stabilize the training of generative adversarial networks. Lewandowski et al. [2025] introduced spectral regularization for continual learning, aiming to prevent the loss of plasticity and maintain trainability across tasks by keeping the maximum singular value of each layer close to one. Regarding SA specifically, Noci et al. [2022] analyzed Jacobians to explain rank collapse, while Castin et al. [2024] evaluated their spectral properties mathematically. In this work, we use Jacobian analysis to understand inference dynamics in realistic SAs and also employ them as regularizers and performance indicators.

**Looped architectures.**  Looped architectures in Transformers have been explored since their introduction by Dehghani et al. [2018]. One example is weight tying, as seen in the ALBERT model [Lan et al., 2020]. Equilibrium models [Bai et al., 2019] use fixed-point solutions, which can be interpreted as infinitely looped computations. Yang et al. [2024], Giannou et al. [2023] showed that Transformers with looped structures are capable of learning algorithmic tasks. Saunshi et al. [2025] further showed that looped architectures enhance reasoning ability through strong inductive bias. As the number of recurrent updates (i.e., loops) increases, performance scales efficiently, a phenomenon we refer to as test-time scaling. Geiping et al. [2025] successfully applied test-time scaling to reasoning benchmarks, and Bansal et al. [2022] showed that it enables models to solve problems at test time that are more difficult than those seen during training. Miyato et al. [2025] proposed artificial Kuramoto oscillatory neurons (AKOrN), a looped architecture that successfully solves tasks in a neuroscience-inspired manner, demonstrating strong empirical results in unsupervised object discovery, adversarial robustness, calibrated uncertainty quantification, and reasoning.

## C.2 Details of preliminaries

**Energy-based analysis by Yang et al. [2022]**   Yang et al. [2022] formalized updates of SA using alternating inexact minimization algorithm as:

$$\boldsymbol{X}^{(t+1)} = \text{softmax}_\beta(\boldsymbol{X}^{(t)}\boldsymbol{W}^s\boldsymbol{X}^{(t)\top})\boldsymbol{X}^{(t)}\boldsymbol{W}^s, \tag{110}$$

where $\boldsymbol{W}^s \in \mathbb{R}^{D \times D}$ is a symmetric matrix and $\text{softmax}_\beta$ is a function reweighted with coefficient vector $\beta$.

**Operation on oscillators**   We use $\widetilde{\boldsymbol{X}}_{i,j}$ to refer to the $j$-th oscillator of the $i$-th token of $X$, which is defined as $\widetilde{\boldsymbol{X}}_{i,j} = \boldsymbol{X}_{[i,(j-1)N+1:jN]} \in \mathbb{R}^N$. They are defined as:

$$\widetilde{\text{Omg}^{(\text{osc})}(\boldsymbol{X}^{(t)})}_{i,j} = \Omega_j \widetilde{\boldsymbol{X}}_{i,j}, \quad \widetilde{\text{Proj}_X^{(\text{osc})}(\boldsymbol{Y})}_{i,j} = \left(I_N - \widetilde{\boldsymbol{X}}_{i,j}\widetilde{\boldsymbol{X}}_{i,j}^\top\right)\widetilde{\boldsymbol{Y}}_{i,j}, \quad \widetilde{\Pi^{(\text{osc})}(\boldsymbol{Y})}_{i,j} = \frac{\widetilde{\boldsymbol{Y}}_{i,j}}{\|\widetilde{\boldsymbol{Y}}_{i,j}\|}, \tag{111}$$

AKOrN then uses a readout module to read out patterns independent of the phase.

$$\boldsymbol{C}' = \boldsymbol{g}(\boldsymbol{m}) \in \mathbb{R}^{D \times N}, m_k = \|\boldsymbol{z}_k\|, \boldsymbol{z}_k = \sum_i \boldsymbol{U}_{kij}\widetilde{\boldsymbol{X}}_{i,j} \in \mathbb{R}^{N'}, \tag{112}$$

where $\boldsymbol{U}_{kij} \in \mathbb{R}^{N' \times N}$ is a learned weight matrix, $\boldsymbol{g}$ is a learned function and $k = 1 \cdots DN$.

