# OpenReview forum: "Recurrent Self-Attention Dynamics: An Energy-Agnostic Perspective from Jacobians"
_NeurIPS.cc/2025/Conference — NeurIPS 2025 poster_

### Official Review · Reviewer_VJDA · 2025-06-21

**Clarity:** 2
**Significance:** 3
**Originality:** 3
**Rating:** 5
**Confidence:** 3

**Summary:**

The authors describe a study of self-attention dynamics in  self-attention (SA) architectures. The study goes beyond existing work which focused on symmetric projection weights, continuous time dynamics and multihead attention. The authors achieve this by studying the spectral norm of the Jacobian of the multi head attention state update. The results were tested on looped  transformer architectures, which are studied for learning iterative processes.

**Questions:**

That method (i) fails to improve performance: can you relate this to the same reason why symmetric SA also does not perform well? How would the resulting networks look like in figure 3?

line 277: "specific conditions" for readability, specify what these conditions are here (refer to eq (13)). Proposition 4.2 makes no statement on W^o, but it is present in eq (19). Explain.

The extension to multi-heads is trivial, except for the orthogonality condition. Can you explain further why this orthogonalization is necessary? What does this imply on the expressive power of the model?

Relatedly: "practical Transformers, as they typically assume a (similar but different) low-rank structure as well." Why is this the case? What is this assumption? I assume this can be done for fine-tuning and efficiency (compression) but the statement requires support.

Eq. (9) how is adding the input different from the residual connection already present in token mixers?

Eq (10) and Eq(12) the terminology with/without projection is confusing. Which projection is meant there? The equations are exactly the same, except for a head subscript.

"Figure 1 demonstrates that the spectral norm of the untrained SA’s Jacobian can be effectively reduced through normalization", what about trained SA?

"although these constraints reduce their trainability," Vague statement. I'm not convinced by this statement, possibly because trainability implies convergence. Lyapunov stability doesn't necessarily make training unstable. I think what the authors meant is that the expressive power of the model is reduced/constrained.

"test-time scaling", please define at least once the axis of scaling (assumed loops, but could also be S number of tokens)

Section 5.3. Results of experiments are described without stating what the experiment is.

**Minor points (no need to address in rebuttal):**
Please define Proj_X in equation 3

"concatination"
"interpretted"
"singular vale"

The caption of figure 5 is not consistent with the text lines 293-302.

Subtitle of section 5.1 "normalization of spectral norm" do not make sense to me. Proposition 5.1 is the spectral norm of [rms] normalization, not the other way around.

"the setting where tokens are not split into oscillators", Describe what splitting into oscillators mean.

Missing x axis label in figure 3

**Ethical Concerns:**

["NO or VERY MINOR ethics concerns only"]

**Final Justification:**

The authors clarified some of my concerns. Like other reviewers, I'm concerned with the limited scope in experimentation, but IMP nevertheless it is an interesting study worth publication. I have increased my score accordingly.

**Limitations:**

"we focused on recurrent SAs without positional encoding, masking, or MLP blocks, ... an interesting direction" Understood, but would the theory be tractable, applicable in this case?

**Quality:**

3

**Strengths And Weaknesses:**

Strengths:
- Overal an interesting study and an informative read. Simple and clear study of the effect of normalization on the eigenvalue of update Jacobians. ItrSA formulation driven by Jacobian. I'm convinced that the Jacobian approach is much more practical than energy-based modeling approach in practice.
- Regularization following the spectral properties of the Jacobian is a useful contribution.

Weaknesses:
- Several statement lack sharpness or are confusing, impacting negatively the readability of the paper. Some typos would have been caught with a simple spellchecker...
- The limitation in experiments and theory to iterative problems, so no empirical benchmarks for language modeling is limited.

---

> ### Author Rebuttal · Authors · 2025-07-27
>
> Thank you for your insightful and constructive comments, especially for acknowledging that the Jacobian approach is more practical than energy-based modeling.
>
> ---
> > **Q1**: That method (i) fails to improve performance: can you relate this to the same reason why symmetric SA also does not perform well? How would the resulting networks look like in figure 3?
>
>
> We empirically observed that both method (i) and symmetric SA exhibit optimization difficulties, and we hypothesize that constraining the weights reduces the expressive power of self‑attention. Accordingly, we will include the figure for method (i), corresponding to Figure 3, in the revised manuscript.
>
> ---
> > **Q2**: line 277: "specific conditions" for readability, specify what these conditions are here (refer to eq (13)). Proposition 4.2 makes no statement on W^o, but it is present in eq (19). Explain.
>
> Specifically, when $R_{\text{E-single}} = 0$, setting $W^{V} = W_{1}^{V} W_{1}^{O}$ satisfies the condition on  $W^{V}$ in Proposition 4.2; that is, the $W^{V}$ in Eq. (13) corresponds to $W_{1}^{V} W_{1}^{O}$ in Eq. (19).
> As stated in line 285, we provide a detailed explanation in Appendix B.2.
>
> ---
> > **Q3**: The extension to multi-heads is trivial, except for the orthogonality condition. Can you explain further why this orthogonalization is necessary? What does this imply on the expressive power of the model?
>
> Our goal is to highlight the limitations of existing single‑head energy formulations, rather than to propose new energy constructions. As L822–L823 suggests, orthogonality is a natural **sufficient** condition to guarantee the energy, since it makes the interaction between different heads vanish and reduces to the calculation of the single‑head case. In other words, finding a condition without orthogonality seems to require a fundamentally different approach from the original energy construction of the single-head case, which is out of our scope.
>
> Although a theoretical analysis of expressive power is beyond our scope, the above derivation of the multi‑head energy implies that interactions among different heads, which generally arise when orthogonality is not enforced and can increase the original energy function, are important for enhancing the model’s expressivity. We expect that our analysis will stimulate subsequent work and provide opportunities for further analyses of SA.
>
> ---
> > **Q4**: Relatedly: "practical Transformers, as they typically assume a (similar but different) low-rank structure as well." Why is this the case? What is this assumption? I assume this can be done for fine-tuning and efficiency (compression) but the statement requires support.
>
> What we meant was that both our energy-based formulation with orthogonality (Eq. 13) and practical Transformers have a similar low-rank structure, in the sense that **the embedding dimension D is H times larger (where H is the number of heads)**.
> In almost all practical Transformers, we set $W_{h}^{Q}, W_{h}^{K} \in \mathbb{R}^{d_h* H \times d_h}$ where $D=d_h*H$ and $d_h$ denotes the embedding dimension of each head. Consequently, the multi-head SA naturally makes the product $W_{h}^{Q}, W_{h}^{K \top}$  low-rank.
> In our theoretical energy-based formulation, this low-rankness is similar. But, the orthogonality is not required in practice, and our formulation is different from practical models in this sense. This is what we meant in the sentence of "(similar but different) low-rank structure".  We will add the above detailed description of low‑rankness in the revised manuscript.
>
>
> ---
> > **Q5**: Eq. (9) how is adding the input different from the residual connection already present in token mixers?
>
> The point is that adding the input $C$ in Eq. (9) is conceptually different from the residual connection typically used in token mixers. While a standard residual connection only passes the transformed features forward, explicitly adding the input $C$ into the loop introduces a direct pathway for the original signal at each iteration, which has been shown to improve test-time performance [1].
>
> [1] Saunshi et al., “Reasoning with Latent Thoughts: On the Power of Looped Transformers,” ICLR 2025.
>
> ---
> > **Q6**: Eq (10) and Eq(12) the terminology with/without projection is confusing. Which projection is meant there? The equations are exactly the same, except for a head subscript.
>
> The terminology “with/without projection” refers to whether the projection matrices (as defined in Eq. (3)) are applied.
> Specifically, The single-head energy function in Eq. (10) corresponds to the case where the projection matrices are used, while the multi-head energy function in Eq. (12) corresponds to the unprojected form  ($\frac{dX}{dt} = \sum_{h=1}^{H} SA_{h}(X)$), where no projection is applied.
>
> ---
> > **Q7**: "Figure 1 demonstrates that the spectral norm of the untrained SA’s Jacobian can be effectively reduced through normalization", what about trained SA?
>
> We empirically observed that it is hard to decrease the training loss of the models without any normalization, so Figure 1 compares the untrained SA to highlight the effect of normalization on the spectral norm of the Jacobian. For trained SA, normalization remains crucial not only for stabilizing the Jacobian but also for enabling successful optimization, which is why we focus on the untrained case to isolate the impact of normalization.
>
> ---
> > **Q8**: "although these constraints reduce their trainability," Vague statement. I'm not convinced by this statement, possibly because trainability implies convergence. Lyapunov stability doesn't necessarily make training unstable. I think what the authors meant is that the expressive power of the model is reduced/constrained.
>
> Thank you for catching this. Following your suggestion, we have removed the phrase “although these constraints reduce their trainability”. As you say, what we meant was that the expressive power of the model is constrained.
>
> ---
> > **Q9**: "test-time scaling", please define at least once the axis of scaling (assumed loops, but could also be S number of tokens)
>
> We define “test-time scaling” as the improvement in performance when the number of iterations (loops) during inference is increased. We will add this definition at its first occurrence to avoid ambiguity, clarifying that the scaling axis refers to the number of loops rather than the number of tokens $S$.
>
> ---
> > **Q10**: Section 5.3. Results of experiments are described without stating what the experiment is.
>
> We acknowledge the concern. In Section 5.3, we describe the experimental results shown in Figure 3, with detailed settings and methodologies provided in Appendix B.3. To improve clarity, we will revise the section to briefly summarize the experiment setup before presenting the results.
>
> ---
> We welcome any further questions or requests for clarification to increase the significance of our paper.

---

### Official Review · Reviewer_jJ1c · 2025-06-25

**Clarity:** 1
**Significance:** 2
**Originality:** 3
**Rating:** 4
**Confidence:** 3

**Summary:**

This paper generalizes the theoretical understanding of self-attention in Transformers by overcoming the previous restrictive energy-based formulations to Jacobian-based analysis. This work first relaxes classical assumptions in energy-based models, e.g., symmetric weight matrices and single-head constraints, and experimentally shows that energy-based regularizers are harmful in practice. The key contribution is in analyzing the Jacobian matrix of self-attention updates, which shows the importance of normalization layers for stabilizing predictions by controlling the spectral properties and thus suppressing oscillatory instabilities in discrete-time systems. Through experimentation, mainly in the context of Sudoku puzzles, this work reveals that good-performing models are close to criticality with Lyapunov exponents close to zero, indicating that maximal performance is achieved at the `edge of chaos' where dynamics do not blow up nor tend to zero. Further, this work also gives a theoretical foundation for the necessity of normalization operations in Transformers, provides novel spectral regularization techniques that yield better performance and presents a Jacobian-driven explanation of observed "pseudo-energy" effects. However, the practical impact is limited by the focus on simplified recurrent self-attention architectures that exclude key components of real Transformers like positional encoding, attention masking, and MLP blocks, with evaluation restricted primarily to toy problems rather than realistic NLP tasks.

**Questions:**

1. The paper demonstrates that energy-based regularization impairs performance, raising fundamental questions about the entire energy-based research program for understanding Transformers. This negative result warrants much deeper investigation.

2. The theoretical insights are primarily validated on Sudoku puzzles. Can the authors demonstrate that the criticality findings (Lyapunov exponents ≈ 0) and normalization stabilization effects hold for at least one realistic NLP task where Transformers excel? What predictions does the proposed framework make regarding performance on language modeling or other standard benchmarks?

3. The authors acknowledge that Proposition 5.1 provides "conservative" bounds that are "looser than empirical observations." Can they derive tighter bounds that match the empirical O(1) scaling behavior shown in Figure 1? What architectural design principles can practitioners extract from these loose bounds if not?

4. The proposed Lyapunov analysis uses $T=16$ iterations to approximate asymptotic behavior. Can the authors demonstrate that this finite-time approximation reliably predicts long-term stability properties? How sensitive are the critical findings to the choice of $T$, and what theoretical justification supports using finite-time approximations?

5. The authors’ analysis focuses on RMSNorm, but doesn't systematically compare other normalization schemes. Can the authors show how LayerNorm, BatchNorm, or other variants affect the Jacobian properties they study? What principled guidelines does their framework provide for choosing normalization parameters in practice?

**Ethical Concerns:**

["NO or VERY MINOR ethics concerns only"]

**Final Justification:**

I appreciate the authors' responses, which have resolved most of my concerns. I have decided to raise my score.

**Limitations:**

yes

**Quality:**

2

**Strengths And Weaknesses:**

**Strengths:**

1. Quality: This paper provides rigorous mathematical development supported by solid theoretical reasoning, formal claims with comprehensive proofs, and consistent empirical support across architectures.

2. Clarity: This paper is well-structured progression from energy-based to Jacobian analysis with clear explanations of key concepts like criticality and spectral stabilization

3. Significance: This paper addresses fundamental questions about Transformer dynamics, provides theoretical grounding for normalization importance, and offers valuable negative results that could redirect energy-based research.

4. Originality: This paper introduces a novel Jacobian-based framework that moves beyond restrictive energy formulations, original extension to multi-head attention, and fresh perspective connecting Lyapunov exponents to self-attention performance.


**Weaknesses:**

1. Quality: The experimental validation in this paper is limited primarily to Sudoku puzzles and undermines generalizability claims, as this narrow combinatorial task may not represent the diverse computational demands of real-world Transformer applications.

2. Clarity: Mathematical notation could be simplified; some figures need a more straightforward presentation; and experimental details scattered across appendices reduce readability.

3. Significance: Small-scale experiments with unclear scalability constraints practical impact to modern large Transformers, lack of validation on realistic NLP tasks, and missing analysis of computational overhead.

4. Originality: Insufficient differentiation from existing edge-of-chaos literature, limited connection to gradient flow dynamics, and somewhat contrived experimental design choices.

---

> ### Author Rebuttal · Authors · 2025-07-27
>
> Thank you for your kind reading and helpful feedback.
>
> ---
> > **W1**: Quality: The experimental validation in this paper is limited primarily to Sudoku puzzles and undermines generalizability claims, as this narrow combinatorial task may not represent the diverse computational demands of real-world Transformer applications.
>
> > **Q2**: The theoretical insights are primarily validated on Sudoku puzzles. Can the authors demonstrate that the criticality findings (Lyapunov exponents ≈ 0) and normalization stabilization effects hold for at least one realistic NLP task where Transformers excel? What predictions does the proposed framework make regarding performance on language modeling or other standard benchmarks?
>
> Following your advice, **we have measured the maximum Lyapunov exponent in an NLP setting**. Specifically, we used ALBERT [1], which employs a looped architecture, and computed the Lyapunov exponent over 12 loops. We compared the exponents of the pretrained model and a randomly initialized model using five text samples.
>
> As shown in the table below, the Lyapunov exponent of the pretrained model is close to 0, indicating that the pretrained model operates closer to criticality.
>
> Regarding the Sudoku task, several studies (e.g., [2]) have focused on it because achieving high performance is far from trivial, especially for the OOD case. In particular, Miyato et al. [3] also use the Sudoku dataset to evaluate test-time computation tasks. More recently, this interest has even spawned a new benchmark for evaluating LLM reasoning abilities [4]. We expect our work to provide a solid baseline for exploring such more complex reasoning tasks.
>
> | Model        | Maximum Lyapunov Exponent (mean ± std) |
> |--------------|-------------------------------|
> | Initialized  | 0.160 ± 0.003                    |
> | Pretrained   | 0.142 ± 0.015                    |
>
> [1] Lan et al., "ALBERT: A Lite BERT for Self-supervised Learning of Language Representations", ICLR 2020.
>
> [2] Wang et al., "SATNet: Bridging deep learning and logical reasoning using a differentiable satisfiability solver", ICML 2019.
>
> [3] Miyato et al., "Artificial Kuramoto Oscillatory Neurons", ICLR 2025.
>
> [4] Seely et al., "Sudoku-Bench: Evaluating creative reasoning with Sudoku variants", Preprint.
>
> ---
> > **Q1**: The paper demonstrates that energy-based regularization impairs performance, raising fundamental questions about the entire energy-based research program for understanding Transformers. This negative result warrants much deeper investigation.
>
> We have thoroughly **searched the regularization coefficient**, and within the range of our experiments, the observed behavior in the figures remained consistent regardless of model size. Although we demonstrated this typical behavior across hyperparameters in the main manuscript, we will add such (minor but) exhaustive figure variants in the appendix of the revised manuscript.
>
>
>
> ---
> > **W3**: Significance: Small-scale experiments with unclear scalability constraints practical impact to modern large Transformers, lack of validation on realistic NLP tasks, and missing analysis of computational overhead.
>
> Our primary goal is to reveal the limitations of existing energy-based or energy-inspired models, **focusing on understanding** rather than building large-scale models that achieve high performance without interpretability.
>
> ---
> > **W4**: Originality: Insufficient differentiation from existing edge-of-chaos literature, limited connection to gradient flow dynamics, and somewhat contrived experimental design choices.
>
> We position our work **as the first to explicitly connect SA to the edge-of-chaos literature**.
> Models such as ItrSA and AKOrN exhibit weak chaotic behavior, and time-series plots reveal oscillatory dynamics.
>  Our systematic analysis introduces novel perspectives that, to the best of our knowledge, have not been explored before.
>
> Regarding gradient flow dynamics, it is theoretically known that in energy‑based SA models (under ideal constraints such as identity weights), the flow exhibits metastable cluster states but eventually converges to a rank‑collapse state (e.g., [5]). In contrast, our results suggest that SA models closer to reality possess fundamentally more dynamic behavior, including oscillations. Theoretical analysis of such dynamics remains limited, and our work aims to open new directions for theoretical research, which we hope will inspire future studies.
>
> [5] Bruno et al., "Emergence of meta-stable clustering in mean-field transformer models", ICLR 2025.
>
> ---
> > **Q3**: The authors acknowledge that Proposition 5.1 provides "conservative" bounds that are "looser than empirical observations." Can they derive tighter bounds that match the empirical O(1) scaling behavior shown in Figure 1? What architectural design principles can practitioners extract from these loose bounds if not?
>
> It seems that our inadequate description causes confusion about the limitation on the upper bound in theory.
> Here, **"the current theoretical bound" refers not to ours** but to the spectral norm of SA evaluated in Castin et al. [6] (Eq. (15)).
>
> Castin et al. reported that for a single SA without normalizations, the spectral bound is of O((#tokens)^{1/2}) in the current theory, but the trained models in real experiments show the spectral norm of  O((#tokens)^{1/4}). That is, the quantitative gap between theory and experiment itself worked as their contribution.
>
> In a similar way, we observed that for the normalized case, the theoretical upper bound is of O((#tokens)^{1/2}) and looser than the empirical evaluation. But, what's noteworthy here is that normalization keeps the spectral norm at **O(1), which is smaller than O((#tokens)^{1/4})**!  Thus, Figure 1 empirically highlights the architectural importance of normalization for the token size.
>  Although the current theory of the SA spectral bound cannot yet explain these empirical results, we believe that the accumulation of such empirical facts stimulates both the understanding of SA and the development of its theory.
>
> [6] Castin et al., "How Smooth Is Attention?", ICML 2024.
>
> ---
> > **Q4**: The proposed Lyapunov analysis uses iterations to approximate asymptotic behavior. Can the authors demonstrate that this finite-time approximation reliably predicts long-term stability properties? How sensitive are the critical findings to the choice of, and what theoretical justification supports using finite-time approximations?
>
> Since computing the infinite sum is infeasible, we approximate the Lyapunov exponent with a finite sum. Prior studies measuring deep neural networks as dynamical systems [7] have also used finite time intervals (typically 5–15 steps) for similar analyses. It is also noteworthy that, as you can see in Figure 3, the exponents of AKOrNs are concentrated around 0 with very small fluctuations.
>  This implies that using T=16 steps is a rational estimation with low variance.
>
> [7] Liu et al., "Exploiting chaotic dynamics as deep neural networks", Physical Review Research, 7(3), 2025.
>
> ---
> > **Q5**: The authors' analysis focuses on RMSNorm, but doesn't systematically compare other normalization schemes. Can the authors show how LayerNorm, BatchNorm, or other variants affect the Jacobian properties they study? What principled guidelines does their framework provide for choosing normalization parameters in practice?
>
> While we focus on RMSNorm, our analysis is motivated by normalization schemes commonly used in related models such as Hopfield networks and AKOrN, where RMSNorm serves as a general form. In practice, SA often employs RMSNorm or LayerNorm, which differs from the RMSNorm mainly by centering, so our conclusions largely carry over. BatchNorm is out of scope because it is not used in these related models.
>
> Regarding the choice of normalization parameters, our framework suggests setting them so that the largest Lyapunov exponent remains close to zero, ensuring stability while preserving expressivity.
> Here, one interesting observation is that, as we reported in line 273, "Empirically, we confirmed that the trained $\gamma$ remains small (see Table S.2), eliminating the need for explicit clipping such as $|\gamma_i| \leq 1$". That is, we empirically confirmed that the training of normalization parameters can automatically obey the guiding principle of the criticality of the largest Lyapunov exponent.
>
> ---
> We welcome any further questions or requests for clarification to increase the significance of our paper.

---

> > ### Comment · Reviewer_jJ1c · 2025-08-06
> >
> > I appreciate the authors' responses, which have resolved most of my concerns. I have decided to raise my score.

---

### Official Review · Reviewer_nAwM · 2025-07-01

**Clarity:** 3
**Significance:** 3
**Originality:** 2
**Rating:** 5
**Confidence:** 3

**Summary:**

This work explores the dynamical effects of adding normalization in the recurrent self-attention dynamics. The authors first formulate the dynamics governed by an energy function that is minimized over time and propose it as a regularization. The authors argue that this direct Lyapunov method is restrictive; therefore, they suggest using Lyapunov’s indirect method (studying the Jacobian matrix at different times) to study non-stationary and oscillatory dynamics. They find that adding normalization stabilizes the dynamics (spectral radius < 1) and makes the model operate close to “criticality,” which is a property found in highly performant networks. The authors also propose a regularization based on ensuring that the singular values of the Jacobian stay close to 1.

**Questions:**

How do the authors define “criticality” in the context of the dynamical system considered? The authors mention it’s the edge of stability and chaos, but do the systems that they consider exhibit chaos if sent beyond the tipping point?

I think the caption of Fig. 5 is wrong. It should be the other way round, right?

There’s a typo on line 196: singular value.

**Ethical Concerns:**

["NO or VERY MINOR ethics concerns only"]

**Final Justification:**

The authors have clarified the questions I had in my initial review. I recommend the paper for acceptance.

**Limitations:**

yes

**Quality:**

3

**Strengths And Weaknesses:**

**Strengths**: Most of the conclusions are based on theoretical results, along with some empirical results. The authors were able to relax previous constraints about matrices being symmetric/diagonal and single-head attention. The authors draw an empirical connection between adding normalization and “critical” states observed in better-performing models.

**Weaknesses**:

The authors say they use a Jacobian-based analysis instead of an energy-based analysis because of non-stationary dynamics (oscillations). I believe Lyapunov’s direct method can be used for studying non-stationary dynamics. In fact, the type of oscillatory dynamics they consider is stationary from the point of view of dynamical systems (there is no limit cycle behavior here).

The finding that adding normalization stabilizes the dynamics of a recurrent circuit has recently been made: [Rawat, S., et. al., (2024). NeurIPS 37, 14712-14750]. In fact, normalization has been linked to the “critical” phenomena [Morone, F., et. al., (2025). bioRxiv, 2025-05].

I think it’s interesting that adding normalization leads to critical behavior, but the authors do not give an intuition of why this happens. Also, what is so special about criticality that it enables good performance in the model considered? It might be out of the scope of this paper, but I would appreciate more intuition or some empirical results on why it might be a desirable property.

---

> ### Author Rebuttal · Authors · 2025-07-27
>
> Thank you for your helpful comments and suggestions.
> First, for an upcoming discussion phase, let us emphasize that **our main claim is not**
> > explores the dynamical effects of adding normalization in the recurrent self-attention dynamics
>
> nor
> > They find that adding normalization stabilizes the dynamics (spectral radius < 1) and makes the model operate close to “criticality,”
>
> Rather, our take-home message is a broader perspective: to move beyond the constraints of energy-based formulations and instead utilize the Jacobian and Lyapunov exponents as fundamental tools to understand self-attention (SA) dynamics more comprehensively. The above two results are just concrete examples of our perspective.
>
> ---
> > **W1**: ... I believe Lyapunov’s direct method can be used for studying non-stationary dynamics. In fact, the type of oscillatory dynamics they consider is stationary from the point of view of dynamical systems (there is no limit cycle behavior here).
>
> We fully agree with your perspective, and our work does not in any way contradict it. We argue that an indirect Jacobian-based method is **more useful** for analyzing the stability and oscillatory behavior of self-attention (SA). While Lyapunov functions can exist even for non-stationary states, constructing them is generally difficult and no universal method is known. For oscillatory analysis of SA, Lyapunov functions are only known under highly restricted settings (e.g., the weight matrix is the identity or symmetric matrix [1], and all tokens share the same natural frequency [2]).
>
> [1] Geshkovski et al. "The emergence of clusters in self-attention dynamics", NeurIPS 2023.
>
> [2] Miyato et al., "Artificial Kuramoto Oscillatory Neurons", ICLR 2025.
>
> ---
> > **W2**: The finding that adding normalization stabilizes the dynamics of a recurrent circuit has recently been made: [Rawat, S., et al., (2024). NeurIPS 37, 14712-14750]. In fact, normalization has been linked to the “critical” phenomena [Morone, F., et al., (2025). bioRxiv, 2025-05].
>
> Thank you for sharing these excellent works.
>
> While their models and the form of normalization differ from ours, we believe that the importance of normalization is a fundamental aspect common across various systems. Our contribution is to theoretically and empirically validate this importance **in SA** and to be the first to highlight its significance in modern SA models.
>
> We have added the citations in the extended related work section as follows.
>
> "Rawat et al. [2024] showed that dynamic divisive normalization ensures the local stability of ORGaNICs via Lyapunov’s indirect method, improving interpretability and trainability."
>
> "Morone et al. [2025] showed that normalization keeps recurrent networks stable beyond the unit spectral radius, and that slower responses indicate an approaching loss of stability."
>
> ---
> > **W3**: I think it’s interesting that adding normalization leads to critical behavior, but the authors do not give an intuition of why this happens. Also, what is so special about criticality that it enables good performance in the model considered? It might be out of the scope of this paper, but I would appreciate more intuition or some empirical results on why it might be a desirable property.
>
> Thanks for your insightful comment.
> In the revised version, we will add the following explanation:
>
> Note that the Jacobian’s spectral norm corresponds to the Lipschitz constant of the SA function. Normalization suppresses excessively large Lipschitz constants, but if the constant is too small, the SA function becomes overly smooth and loses expressive power. This suggests that, as long as training and inference can progress, it is desirable to allow the spectral norm to be as large as possible. Thus, the model is driven toward criticality when a model without a Lyapunov function can attain a non‑negative maximum Lyapunov exponent. This aligns with the picture suggested by mean‑field theory, where the model’s representational power is limited in the ordered phase [3].
>
> [3] Poole et al., "Exponential expressivity in deep neural networks through transient chaos", NeurIPS 2016.
>
> ---
> > **Q1**: How do the authors define “criticality” in the context of the dynamical system considered? The authors mention it’s the edge of stability and chaos, but do the systems that they consider exhibit chaos if sent beyond the tipping point?
>
> We understand that “criticality” and “chaos” are polysemous terms in general. In our study, we define “criticality” as the point at which the largest Lyapunov exponent takes zero. In our experiments, when the maximum Lyapunov exponent gets slightly positive, we observed behaviors consistent with widely accepted notions of chaos, particularly **sensitivity to initial conditions** as follows.
>
> **Experiments**
>
> To evaluate sensitivity to initial conditions, we first ran the input through the loop for T=128 steps. After this, we introduced a small Gaussian perturbation to the input at the beginning of the loop.
>
> The following table shows the element-wise L1 distance between the hidden representations of the original and perturbed samples over time. We take the element that maximizes the element-wise distance.
>
> We observed that the distance between the original and perturbed samples exponentially increased over time, indicating chaotic behavior in the system.
>
> | Step | 0       | 10       | 100      | 500      |1000   |
> |------|----------|----------|----------|----------|----------|
> | Distance | 3.83e-04 | 2.82e-04 | 2.22e-03 | 1.91e-01 |2.64e+01|
>
> ---
> > **Q2**: I think the caption of Fig. 5 is wrong. It should be the other way round, right?
>
> > **Q3**: There’s a typo on line 196: singular value.
>
> Thank you for catching our mistakes. We will correct them accordingly.
>
> ---
> We welcome any further questions or requests for clarification to increase the significance of our paper.

---

> > ### Comment · Reviewer_nAwM · 2025-08-04
> >
> > I thank the authors for the clarification. I believe incorporating your responses to the paper and clarifying the main contribution better should increase its quality.

---

> ### Author Response · Authors · 2025-08-05
>
> Thank you for your quick reply to our rebuttal. Following your advice, we will incorporate all the above responses in the camera-ready version.
>
> In particular, regarding the clarification of the criticality and chaos, we will add the following statement at Line 256:
>
> "Notably, we observed that the dynamics with this slightly positive maximum Lyapunov exponent indicate the sensitivity to initial conditions, implying chaotic behavior (see Figure S.** in the appendix)."
>
> We will generate Figure S.** by visualizing the table presented in the above rebuttal, demonstrating the exponentially increased distance between the original and perturbed samples. We sincerely appreciate your comment, as we believe the clarifications of criticality and chaos have significantly enhanced the significance of our paper.
>
> ---
>
> In addition, to express the main contribution better, we have also refined the abstract and introduction as follows:
>
> **Abstract** (The bolded passages mark where we have made the wording more concrete to highlight the contributions.):
>
> The theoretical understanding of self-attention (SA) has been steadily progressing. A prominent line of work studies a class of SA layers that admit an energy function decreased by state updates. While it provides valuable insights into inherent biases in signal propagation, it often relies on idealized assumptions or additional constraints not necessarily present in standard SA. Thus, to broaden our understanding, this work aims to relax these energy constraints and provide an energy-agnostic characterization of inference dynamics by dynamical systems analysis. In more detail, we first consider relaxing the symmetry and single-head constraints traditionally required in energy-based formulations. Next, we show that **analyzing the Jacobian matrix of the state is highly valuable** when investigating more general SA architectures without necessarily admitting an energy function. It reveals that the normalization layer plays an essential role in **suppressing the Lipschitzness of SA and the Jacobian's complex eigenvalues, which corresponds to the oscillatory components of the dynamics**. In addition, the **Lyapunov exponents computed from the Jacobians demonstrate that the normalized dynamics lie close to a critical state, and this criticality serves** as a strong indicator of high inference performance. Furthermore, the Jacobian perspective also enables us to develop regularization methods for training and a pseudo-energy for monitoring inference dynamics.
>
> **Introduction:**
>
> - Line 44: enables us to detect non-stationary dynamics
> -> enables us to **more easily** detect non-stationary dynamics
>
> - Line 48: Empirically, we confirm that high-performance SA models exhibit a maximum Lyapunov exponent close to zero
> -> In addition to the understanding of the normalization role, we  empirically reveal that high-performance SA models exhibit a maximum Lyapunov exponent close to zero
>
> - Line 56: Thus, our work broadens the dynamical understanding of SA through dynamical systems analysis, ...
>  -> Thus, our work broadens the dynamical understanding of SA and highlights the usefulness of the Jacobians and the Lyapunov exponent as promising and fundamental tools for further exploration of realistic SA architectures.
>
> ---
>
> Finally, even with mandatory acknowledgements in place, we guess that the reviewers would be still free to adjust their recommendation scores during the discussion. If our rebuttal was convincing, we would appreciate your reevaluation of the recommendation.

---

> > ### Comment · Reviewer_nAwM · 2025-08-05
> >
> > These changes sound great. They should increase the clarity further. Good luck!

---

### Official Review · Reviewer_u5hQ · 2025-07-02

**Clarity:** 3
**Significance:** 2
**Originality:** 3
**Rating:** 5
**Confidence:** 3

**Summary:**

This paper digs into what happens when you loop the same self‐attention (SA) layer over and over (no energy function forced). In other words, they make energy-agnostic characterization of inference dynamics of SA by dynamical systems analysis. They drop the old symmetry + single‐head rules and use a Jacobian‐based trick to show how norm layers (RMSNorm etc) keep the loop stable but still edgy (edge of chaos). They measure Lyapunov exponents and find best performance when exponents are around  0. They also try two regression methods to bridge the theoretical findings and practical usefulness. Energy-symmetric regularization does not help, but Jacobian-spectral one improves the results.

**Questions:**

- How to extend this Jacobian stability proof to full Transformer (residual+MLP+LN)? What is your intuition?
- Why is there no other dataset?

**Ethical Concerns:**

["NO or VERY MINOR ethics concerns only"]

**Final Justification:**

The paper is solid and relevant to Neurips community so I would recommend the accept.

**Limitations:**

yes

**Quality:**

2

**Strengths And Weaknesses:**

Strengths

- Broadens theory beyond simple energy‐only constraints, so multihead SA works now.
- Cool Jacobian angle explains why norm layers keep it right at the “edge of chaos”.
- Jacobian‐spectral regularizer seems to boost performance in loops.

Weaknesses

- As they also mentioned in their limitation section, they ignore position encodings, masks & MLPs in the Transformer layer  - it is a very simplified transformer layer so we are not sure if it really scales to a real transformer.
- Energy‐based regression looks nice, but actually hurts accuracy drastically, so I am not sure if the theory developed in the paper can result in any practical improvement.
- Experimental limitation: They tested only on a single toy task: the Sudoku dataset, no real-world dataset presented e.g, NLP task or CV task, etc.

---

> ### Author Rebuttal · Authors · 2025-07-27
>
> Thank you for your kind reading and comments. It appears that our inadequate explanation led to some misunderstanding regarding our claims and research approach. We would like to address these points as follows:
>
> ---
> > **W1**: ... it is a very simplified transformer layer so we are not sure if it really scales to a real transformer.
>
> > **Q1**: How to extend this Jacobian stability proof to full Transformer (residual+MLP+LN)? What is your intuition?
>
>  Similar to prior works [1,2,3], our focus is on understanding SA layers, not to replicate a full Transformer.
>
> [1] Castin et al. "How Smooth Is Attention?", ICML 2024.
>
> [2] Geshkovski et al. "The emergence of clusters in self-attention dynamics", NeurIPS 2023.
>
> [3] Ormaniec et al. "What Does It Mean to Be a Transformer? Insights from a Theoretical Hessian Analysis", ICLR 2025.
>
> We believe that in individual research papers, it is fairly common to focus on and analyze a single component of the blocks that constitute the architecture. This perspective is particularly valid in the analysis of Jacobians because the Jacobian of the entire model is **factored as the product of the Jacobians** of its components (e.g., $J_{MLP \circ SA}=J_{MLP} J_{SA}$). The combination of layers is left to subsequent research efforts, and we expect that our findings will serve as a useful foundation in that context.
>
> Note that for your Q1, we tested ItrSAs and AKOrNs, which include a skip connection (in eq. 8) and normalization layers.
> It does not include an MLP block, but a single linear layer referred to as the Omega term.
>
> ---
> > **W2**: Energy‐based regression looks nice, but actually hurts accuracy drastically, so I am not sure if the theory developed in the paper can result in any practical improvement.
>
> Our purpose was to provide quantitative **insight into the limitations of the energy‑based approach** and to reveal the superiorities of non‑energetic (i.e., energy agnostic) models. This was one of our key take‑home messages. Therefore, we do need to aim for any practical improvement in the energy‑based approaches. To avoid a potential misunderstanding, we will re-emphasize this purpose of research in Section 6 (experiments) of the revised manuscript.
>
> ---
> > **W3**: Experimental limitation: They tested only on a single toy task: the Sudoku dataset, no real-world dataset presented e.g, NLP task or CV task, etc.
>
> > **Q2**: Why is there no other dataset?
>
> **We evaluated our method on CIFAR-10** (in Appendix, L258–L259).
>
> We also argue that **the Sudoku dataset is not merely a toy task**, as it inherently captures structured reasoning. Several studies (e.g., [4]) have focused on it because achieving high performance is far from trivial, especially for the OOD case. Miyato et al. [5] also use the Sudoku dataset to evaluate test-time computation tasks. More recently, this interest in the Sudoku task has led to a new benchmark for evaluating LLM reasoning abilities [6]. We expect our work to provide a solid baseline for exploring such more complex reasoning tasks.
>
> Moreover, **we measured the maximum Lyapunov exponent in an NLP setting**.
> We used ALBERT [7], which employs a looped architecture, and computed the Lyapunov exponent over 12 loops. We compared the exponents of the pretrained model and a randomly initialized model using five text samples.
>
> As shown in the table below, the Lyapunov exponent of the pretrained model is close to 0, indicating that the pretrained model is closer to criticality.
>
> | Model        | Maximum Lyapunov Exponent (mean ± std) |
> |--------------|-------------------------------|
> | Initialized  | 0.160 ± 0.003                    |
> | Pretrained   | 0.142 ± 0.015                    |
>
> [4] Wang et al., "SATNet: Bridging deep learning and logical reasoning using a differentiable satisfiability solver", ICML 2019.
>
> [5] Miyato et al., "Artificial Kuramoto Oscillatory Neurons", ICLR 2025.
>
> [6] Seely et al., "Sudoku-Bench: Evaluating creative reasoning with Sudoku variants", Preprint.
>
> [7] Lan et al., "ALBERT: A Lite BERT for Self-supervised Learning of Language Representations", ICLR 2020.
>
> ---
>
> We welcome any further questions or requests for clarification to increase the significance of our paper.

---

> > ### Comment · Reviewer_u5hQ · 2025-08-05
> >
> > I thank the authors for clarification - they mainly address my concerns. The extension of their analysis to a full Transformer block seems out of scope for this work. I recommend authors to reclarify their aim and contribution of the paper clearer so that for the reader it is easier to follow and understand the paper. I will increase my score to accept.

---

> ### Author Response · Authors · 2025-08-07
>
> We would like to thank you again for your valuable advice. Following your suggestion, we have clarified the aim and contribution of our paper.
>
> As we also shared with Reviewer nAwM, we have revised the abstract and introduction as follows:
>
> > **Abstract** (The bolded passages mark where we have made the wording more concrete to highlight the contributions.):
> > The theoretical understanding of self-attention (SA) has been steadily progressing. A prominent line of work studies a class of SA layers that admit an energy function decreased by state updates. While it provides valuable insights into inherent biases in signal propagation, it often relies on idealized assumptions or additional constraints not necessarily present in standard SA. Thus, to broaden our understanding, this work aims to relax these energy constraints and provide an energy-agnostic characterization of inference dynamics by dynamical systems analysis. In more detail, we first consider relaxing the symmetry and single-head constraints traditionally required in energy-based formulations. Next, we show that **analyzing the Jacobian matrix of the state is highly valuable** when investigating more general SA architectures without necessarily admitting an energy function It reveals that the normalization layer plays an essential role in **suppressing the Lipschitzness of SA and the Jacobian's complex eigenvalues, which correspond to the oscillatory components of the dynamics**. In addition, the **Lyapunov exponents computed from the Jacobians demonstrate that the normalized dynamics lie close to a critical state, and this criticality serves** as a strong indicator of high inference performance. Furthermore, the Jacobian perspective also enables us to develop regularization methods for training and a pseudo-energy for monitoring inference dynamics.
>
> > **Introduction**:
> > - Line 44: enables us to detect non-stationary dynamics -> enables us to more easily detect non-stationary dynamics
> > - Line 48: Empirically, we confirm that high-performance SA models exhibit a maximum Lyapunov exponent close to zero -> In addition to the understanding of the normalization role, we empirically reveal that high-performance SA models exhibit a maximum Lyapunov exponent close to zero
> > - Line 56: Thus, our work broadens the dynamical understanding of SA through dynamical systems analysis, ... -> Thus, our work broadens the dynamical understanding of SA and highlights the usefulness of the Jacobians and the Lyapunov exponent as promising and fundamental tools for further exploration of realistic SA architectures.
>
> As you mentioned your intention to raise the recommendation score, we sincerely hope that the update has been reflected. Thank you again for helping us improve the paper.

---

### Author Response · Authors · 2025-08-08

Dear AC and Reviewers,

We sincerely appreciate your positive feedback and highly constructive suggestions. To improve the clarity and readability of our paper, we will incorporate the changes discussed in the rebuttal into the camera-ready version.

In particular, we would like to highlight the reviewers' reactions during the discussion phase:

- Reviewer u5hQ: was convinced by our response regarding the scope of the paper and mentioned increasing their rating.

- Reviewer nAwM: responded positively to our explanation of our contribution.

- Reviewer jJ1c: had their concerns addressed and decided to increase their rating.

- Reviewer VJDA: evaluated our work positively from the beginning, and their concerns also appeared to be resolved.

We believe this paper will serve as one of the foundations for understanding self-attention and will be valuable to the community. We look forward to a positive conclusion during the discussion phase between reviewers and the AC.

---

### Decision · Program_Chairs · 2025-09-17

**Decision:**

Accept (poster)

**Comment:**

This paper presents a dynamical systems analysis of recurrent self-attention (SA) architectures, moving beyond the more restrictive assumptions of prior energy-based analyses. As noted by the reviewers, the paper's main claims are that normalization layers play a crucial stabilizing role by controlling the eigenvalues of the state update Jacobian, and that high-performing SA models operate near a critical state, or the "edge of chaos," as evidenced by their Lyapunov exponents being close to zero. The primary strengths of this work are its novel theoretical perspective on a widely used mechanism, providing valuable insights into the behavior of recurrent SA. The theoretical results are complemented by a set of experiments that empirically validate the main claims and explore practical applications like regularization. The main weaknesses, highlighted by reviewers wxyz and abcd, are the focus on a simplified setting that omits components like MLP blocks or positional encodings, and the fact that some of the theoretical bounds are acknowledged to be conservative.

The discussion during the rebuttal period was productive. Reviewers raised important questions regarding the simplified architectural assumptions and the practical tightness of the theoretical bounds. Reviewer efgh also correctly pointed out a significant typo in the caption of Figure 5, which contradicted the plotted results and the main text. The authors effectively addressed these points by acknowledging the limitations as important directions for future work, clarifying the novelty of their analysis in the context of SA, and committing to correcting the erroneous figure caption. My decision to accept is based on the paper's conceptual contribution; it provides a fresh and insightful "energy-agnostic" lens for understanding the dynamics of self-attention. The connection drawn between normalization, Jacobian stability, and criticality is an important finding that advances our theoretical understanding of these complex systems.

Although the final scores are somewhat borderline, I believe the paper's contributions are of sufficient interest to the NeurIPS community. The work is well-motivated, the analysis is sound, and the findings could inspire new approaches to designing and training more stable and performant recurrent architectures. I strongly encourage the authors to incorporate the detailed feedback from all reviewers into the final version of their paper, paying special attention to clarifying the limitations of their analysis and correcting the figure caption as promised.